

# Modeling boreal forest soil dynamics with the microbially explicit soil model MIMICS+ (v1.0)

Elin Ristorp Aas[1,3], Heleen A. de Wit[2,3], and Terje K. Berntsen[1,3]

[1]Department of Geosciences, University of Oslo, Oslo, Norway
[2]Norwegian Institute for Water Research, Økernveien 94, 0579 Oslo
[3]Centre for Biogeochemistry in the Anthropocene, Department of Geosciences, University of Oslo, Oslo, Norway

**Correspondence:** Elin Ristorp Aas (ecaas@uio.no)

**Abstract.** Understanding carbon exchange processes between land reservoirs and the atmosphere is essential for predicting carbon-climate feedbacks. Still, considerable uncertainty remains in the representation of the terrestrial carbon cycle in Earth System Models. An emerging strategy to constrain these uncertainties is to include the role of different microbial groups explicitly. Following this approach, we extend the framework of the MIcrobial-MIneral Carbon Stabilization (MIMICS) model

with additional mycorrhizal groups and a nitrogen cycle that includes a novel representation of inorganic nitrogen sorption to particles via a Langmuir isotherm. MIMICS+ v1.0 is designed to capture and quantify relationships between soil microorganisms and their environment, with a particular emphasis on boreal ecosystems. We evaluated MIMICS+ against podzolic soil profiles in Norwegian forests as well as the conventional Community Land Model (CLM). MIMICS+ matched observed carbon stocks better than CLM, and gave a broader range of C:N ratios, more in line with observations. This is mainly explained

by a higher direct-plant-derived fraction into the Soil Organic Matter (SOM) pools. The model produces microbial biomass estimates in line with numbers reported in the literature. MIMICS+ also showed better representation of climate gradients than CLM, especially in terms of temperature. To investigate responses to changes in nutrient availability, we performed an N enrichment experiment, and found that nitrogen sorbed to particles through the sorption algorithm served as a long-term storage of nutrients for the microbes. Furthermore, although the microbial groups responded considerably to the nitrogen enrichment,

we only saw minor responses for carbon storage and respiration. Together, our results present MIMICS+ as an attractive tool for further investigations of interactions between microbial functioning and their (changing) environment.

## 1   Introduction

Among the carbon (C) stores in the terrestrial biosphere soils are the largest, containing ca. 1700 GtC, while vegetation accounts for ca. 450 GtC globally (Friedlingstein et al., 2022). The active exchange of C between terrestrial pools and the atmosphere is

affected by elevated $CO_2$ concentrations and changes in N deposition, but quantifying the responses has proven to be a central challenge in climate science. Arora et al. (2020) highlight the uncertainty in terrestrial carbon-concentration and carbon-climate feedbacks from the last model intercomparison project, CMIP6. The uncertainty of carbon-cycle feedbacks is up to one order of magnitude larger for land than for ocean, illustrating the need to improve model representation of terrestrial processes. To do this, we need to represent complex C and nutrient cycle processes in a modeling framework, a task that requires careful



consideration of how to translate real-world processes into an appropriate model form. Fisher and Koven (2020) suggest an
approach based on modular complexity. Dividing a full-complexity land model into smaller modules allows investigation of
alternatives for structure and parameter choices, which helps in making good modeling choices and thereby constrain sources
of uncertainty.

   Large variations in responses between different biomes introduce an extra challenge to C cycle modeling. The impact of
environmental changes on boreal systems is of particular interest for several reasons. Studies show that the kinetics of soil
microbes accustomed to cooler climates are more temperature sensitive than microbes in warmer climates (German et al.,
2012), and soil carbon turnover times have been shown to be more sensitive to climatological temperature than in warm
climates (Koven et al., 2017). Many boreal areas also experience treeline migration caused by an expansion of the temperature-
limited area tree species can grow (Hansson et al., 2021). Often this leads to a shift in mycorrhizal associations, which again
can lead to changes in soil carbon dynamics and below-ground carbon storage (Taylor et al., 2016; Tonjer et al., 2021). The
dominating soil type in boreal/Norwegian forests is podzol (Strand et al., 2016). These soils are typically nutrient-poor, and
competition for nutrients is expected to be important for the carbon dynamics in these systems. Because of the essential role
of boreal processes for terrestrial C cycling and possible climate feedbacks, it is important to represent these processes well in
models. Despite the importance of boreal systems, many soil model structure and parameter choices are based on temperate or
tropical observations. This bias may skew model results, and making the models unfit to represent responses to climate change
in boreal environments.

   Nitrogen (N) is one of the most important nutrients in an ecosystem, and the cycling of nitrogen between above-ground
and below-ground reservoirs can greatly affect carbon dynamics. In addition to regulating forest productivity, N availability
can regulate microbial Carbon Use Efficiency (CUE), as microbes respire excess C to meet their stoichiometrical demand
(Mooshammer et al., 2014b). This direct relationship between N and atmospheric C exchange emphasize the importance of
including microbial C-N relationships in C cycle models. One factor determining nitrogen availability in an ecosystem is
inorganic N deposition from the atmosphere and agricultural fertilization. This inorganic N is subject to physical and chemical
processes that affect how readily available the N is to microbes and plants. One such process is cation exchange, which
determines the fraction of inorganic N that is sorbed onto clay or organic particles (Bonan, 2015), and therefore are temporarily
unavailable for microbial and plant uptake. This is a process that might be extra important in nutrient-poor boreal forest systems.
There are studies examining this effect in agricultural soils (Sieczka and Koda, 2016), but few are looking at natural soils.

   Traditionally, decomposition processes in models have been represented by first-order kinetics between litter and active,
slow, and passive pools of Soil Organic Matter (SOM) (Parton et al., 1988). This approach limits the ability to examine
the mechanisms and possible responses of the soil system during climate change (Todd-Brown et al., 2012). Newer work
has introduced models that in different ways represent microbial activity explicitly (e.g., Wieder et al., 2015; Sulman et al.,
2019; Fatichi et al., 2019; Yu et al., 2020). These models increase the possibility to capture carbon climate feedbacks of
the future (Tang and Riley, 2014; Hararuk et al., 2015). Wieder et al. (2015) illustrated that by representing the functional
traits of microbes in a C-only version of the MIMICS model, we can raise important hypotheses about how microbes can
determine responses to N enrichment. Kyker-Snowman et al. (2020) further showed that adding an N cycle to the MIMICS





model (MIMICS-CN) produced results in line with measurements from North American sites, and comparable models. Wang et al. (2021) presented a vertically resolved C-only version of MIMICS and showed that microbial activity and root carbon inputs were more important than the soil carbon diffusion when simulating soil carbon concentration profiles.

     Building on the insights from the above-mentioned studies, we expanded the MIMICS framework with a representation of mycorrhizal associations using methods presented by Sulman et al. (2019) and Baskaran et al. (2017). In addition, we

introduced an algorithm for representing sorption of ammonium to soil particles based on the Langmuir isotherm (Sieczka and Koda, 2016), which may be an important but underrepresented process determining the availability of inorganic N to soil microbes in boreal forests. We assume that by including processes and parameters thought to be particularly relevant for climate responses in colder areas we can obtain a better understanding of the C dynamics, and thereby reduce uncertainty connected to soil processes. Still, a future goal is to couple the soil model to a land model/ESM with interactive vegetation. Although our

present emphasis is on boreal systems, the incorporated processes are general and representative on a larger scale. We introduce a vertically resolved, microbially explicit soil decomposition model, MIMICS+, that represents C and N flows between litter, microbial and SOM pools. In this study the model is offline, and forced with data produced by the Community Land Model v5.1 (CLM; Lawrence et al. (2019)). C and N stock estimates from the CLM simulations represent a microbially implicit approach based on the traditional CENTURY model (Parton et al., 1988). Therefore, we compare the CLM and MIMICS+ results to

investigate the implications of including the processes and mechanisms mentioned above. To evaluate the model, we use a collection of soil profile data from forested, podzolic sites in Norway, covering a range of conditions representative of boreal systems (Strand et al., 2016). Our experimental setup is as follows; for a selection of 50 sites in Norway, we ran simulations with the CLM model to produce a) input data needed to run MIMICS+, and b) estimates of C and N stocks. We then ran MIMICS+ with the produced forcing data. The aims of the study are 1) to formulate a standalone, microbially explicit model

capable of representing soil processes in boreal systems, 2) to evaluate model performance and model structure by comparing simulated vertical soil C content along a climatic gradient with observations and simulated soil carbon from the microbially implicit model CLM 3) Apply the model to perform an N enrichment experiment to investigate below-ground responses to nutrient changes.

## 2 Model and Methods

**2.1 Model Description**

   MIMICS+ is based on the MIMICS framework where microbial groups, litter and soil organic matter are represented as separate pools (Wieder et al., 2015). In its current state MIMICS+ is not coupled to a comprehensive land model, and therefore needs prescribed C and N input, and soil temperature and moisture, which it is set up to read from CLM history files. Mass balance equations, $\mathrm{dP/dt = Sources - Sinks}$, determine the change at each timestep for each pool, P. The model structure

with pools and fluxes is illustrated in Fig. 1, and a detailed overview of mass balance and rate equations are provided in the Appendix; Tables A1 and A2 contain mass balance equations for carbon and nitrogen pools, respectively, while Tables A3 and A4 list carbon and nitrogen rate equations. Throughout the model description, fluxes referred to as "CX" or "NX", where X





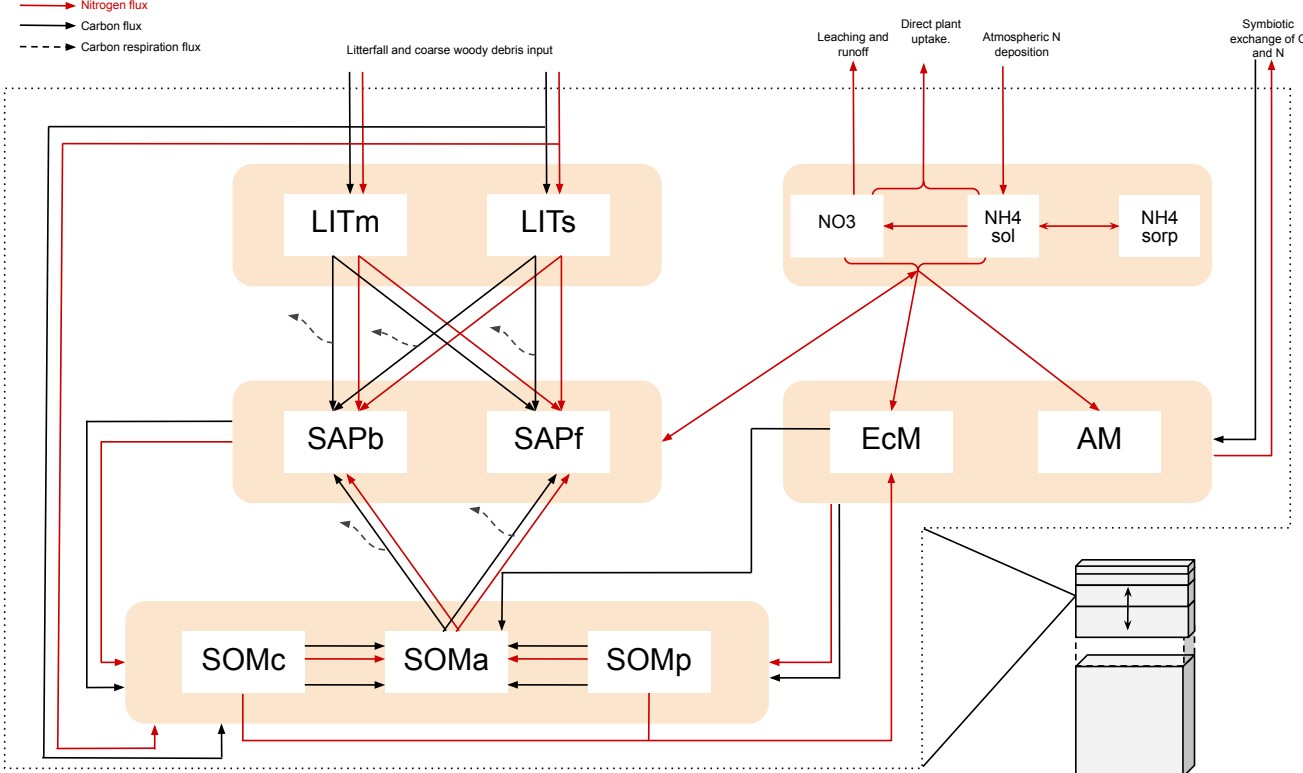

**Figure 1.** Schematic showing C and N flows within each layer of the model. Black arrows indicate carbon fluxes ($\mathrm{gCm^{-3}h^{-1}}$) while red arrows indicate nitrogen fluxes ($\mathrm{gNm^{-3}h^{-1}}$). The dashed black arrows symbolize C leaving the system as heterotrophic respiration. LITm, LITs: metabolic and structural litter. SAPb, SAPf: saprotrophic bacteria and fungi. EcM, AM: ecto- and arbuscular mycorrhizal fungi. SOMc, SOMa, SOMp: chemically protected, available and physically protected soil organic matter. $NO_3$, $NH4_{sol}$, $NH4_{sorb}$: Inorganic N in the form of $NO_3$, $NH_4$ in solution and $NH_4$ sorbed to particles, respectively.

is a number, can be found in those tables, and are illustrated as arrows in Fig. 1 and Fig. A1. A list of parameters is given in Table A5. By representing the same hydrologically and biogeochemically active layers as in CLM, MIMICS+ can represent

the depth discretization of temperature and moisture dependent processes. For each layer the fluxes between the pools within the layer are calculated first, before the vertical transport is calculated. Unless otherwise stated, the equations below describe transport within one layer. The vertical transport is described in Sect. 2.1.4.

### 2.1.1   Litter and SOM pools

Organic C and N enters the litter and SOM pools as dead plant material. As in MIMICS (Wieder et al., 2015) and ORCHIDEE-

SOM (Camino-Serrano et al., 2018), we separate between metabolic (labile) litter mainly originating from leaves and fine roots, and structural litter, in which we also include Coarse Woody Debris (CWD). For SOM we again follow the MIMICS approach





with two protected SOM pools, and one pool that is available for saprotrophic decomposition. Desorption and oxidation move organic matter from physically and chemically protected pools, respectively, to the available pool (C11, C12, N11, N12). 50 % of the incoming metabolic and structural litter goes to physically and chemically protected SOM, respectively, as direct

plant-derived SOM (C3, C4, N3, N4). The direct litter fluxes, together with microbial necromass (C13-C24, N13-N24) and a flux representing EcM enzyme production (C27) are the sources of input to the SOM pools. The microbial pools determine the rates of decomposition, and thereby the transfer rates between the main storage pools; SOM and litter.

### 2.1.2  Microbial processes

MIMICS+ represents two different types of microbes: saprotrophs and mycorrhizal fungi. Within these two main groups we

separate between two functional traits, giving four different microbial pools in total. We divide between saprotrophic fungi (SAPf; analog to MIMICS k-strategists) and bacteria (SAPb; analog to MIMICS r-strategists). Temperature-sensitive reverse Michaelis-Menten kinetics, together with a moisture modifier (Wieder et al., 2017), determines the rates at which saprotrophs decompose substrate from the two litter pools, and the available SOM (C5-C10, N5-N10). The N fluxes are determined by the stoichiometry of the substrate pools. During decomposition, a fraction of the incoming C is lost from the soil as heterotrophic

respiration (HR), while the rest is contributing to saprotrophic biomass. The respired fraction is determined by the carbon use efficiencies $CUE_b$ and $CUE_f$ which have maximum values of 0.4 and 0.7 for bacteria and fungi respectively, but is reduced under low nutrient conditions. This is based on the theory that microbes adjust their efficiencies to maintain a relatively constant, low C:N ratio despite the higher C:N ratio of substrates (Mooshammer et al., 2014b). The C:N ratio of the model saprotrophs is assumed to be constant ($CN_b = 5$ and $CN_f = 8$, Table A5). To ensure that this ratio is fulfilled in each layer and time step

(in addition to potentially reducing CUE) N is exchanged between the saprotrophs and the inorganic pools (N36 and N37). The exchange rates can be positive or negative, leading to either immobilization or mineralization of inorganic N. We first calculate the uptake and demand of N to determine if there is enough to meet the requirement for optimal saprotrophic functioning.

$$N_{demand,x} = \frac{CUE_x \cdot (FC_{LITm,SAPx} + FC_{LITs,SAPx} + FC_{SOMa,SAPx})}{CN_x} \qquad (1)$$

$N_{uptake,x} = NUE \cdot (FN_{LITm,SAPx} + FN_{LITs,SAPx} + FN_{SOMa,SAPx})$ \qquad (2)

Here, x represents either b (bacteria) or f (fungi) and NUE is Nitrogen Use Efficiency, further described below. This results in one of four possibilities:

1. The N demand is greater than the uptake for both bacteria and fungi, meaning both groups will immobilize inorganic N. In this case we check if there is enough available inorganic N to fulfill the demand from both groups. If not, CUE is

reduced (according to Eq. (3) and (4)) so that the saprotrophs utilize all N that is available to them, before the demand is recalculated. Here, $N_{for\_sap}$ is referring to the sum of the available N pools, $N_{NH4,sol}$ and $N_{NO3}$:

$$CUE_b = \frac{(f_b \cdot N_{for\_sap} + N_{uptake,b} \cdot dt) \cdot CN_b}{(FC_{LITm,SAPb} + FC_{LITs,SAPb} + FC_{SOMa,SAPb}) \cdot \Delta t} \qquad (3)$$





$$CUE_f = \frac{((1 - f_b) \cdot N_{for\_sap} + N_{uptake,f} \cdot dt) \cdot CN_f}{(FC_{LITm,SAPf} + FC_{LITs,SAPf} + FC_{SOMa,SAPf}) \cdot \Delta t} \qquad (4)$$

where $f_b$ determines the division of the available inorganic N between bacteria and fungi, and is calculated as:

$$f_b = \frac{(N_{demand,b} - N_{uptake,b})}{((N_{demand,b} - N_{uptake,b}) + (N_{demand,f} - N_{uptake,f}))} \qquad (5)$$

Equation (3) and (4) reduces CUE (and increase the respired fraction) enough to maintain the C:N ratios under the prevailing conditions, and the resulting exchange rates is:

$$FN_{IN,SAPb} = f_b \cdot N_{for\_sap} \qquad (6)$$

$$FN_{IN,SAPf} = (1 - f_b) \cdot N_{for\_sap} \qquad (7)$$

2. N uptake is larger than demand for both saprotrophic groups, meaning both will mineralize inorganic N. The mineralized N will enter the $N_{NH4_{sol}}$ pool.

3. Fungi will mineralize N (uptake > demand), while bacteria immobilizes N (uptake < demand). In this case bacteria can access the N mineralized by fungi in addition to the inorganic N if needed.

4. Bacteria will mineralize N (uptake > demand), while fungi immobilizes N (uptake < demand). In this case fungi can access the N mineralized by bacteria in addition to the inorganic N if needed.

Saprotrophic necromass is transferred to the SOM pools, and is partitioned between the three pools based on clay content of the soil and the metabolic fraction of incoming litter (C13-C18 and N13-N18). Only a fraction of the N released during decomposition is directly available to saprotrophs, determined by the NUE (constant NUE = 0.8, Mooshammer et al. (2014a)). The remaining fraction is transferred to N$_{NH4,sol}$.

The model represent two different types of mycorrhizal fungi: Ectomycorrhiza (EcM) and Arbuscular Mycorrhiza (AM). The mycorrhizal pools receive a C supply from plants, and in return provides N to its associated plants. How the incoming carbon ($I_{veg,Myc}$, cf. C28 and C29) is partitioned between EcM and AM is determined dynamically through a Return Of Investment (ROI) function based on the method from (Sulman et al., 2019). The partition between EcM and AM is determined as a fraction,

$$f_{alloc,i} = \frac{ROI_i}{\Sigma_i ROI_i} \qquad (8)$$

where $ROI_i$ is the nitrogen return of the carbon investment from mycorrhizal association $i$ (EcM or AM);

$$ROI_i = \frac{N_{aquired,i} \cdot \tau_{myc,som} \cdot CUE_i}{C_i} \qquad (9)$$





EcM acquires N from the protected SOM and inorganic N pools ($N_{aquired,EcM} = N25 + N26 + N27$) while AM only acquires inorganic N ($N_{aquired,AM} = N28$). $\tau_{myc,som}$ is the mycorrhizal turnover time, while $CUE_i$ is the growth efficiency for mycorrhizal association $i$. N25 and N26 represent ectomycorrhizal mining for N (Lindahl and Tunlid, 2015). By releasing enzymes (C27) EcM access N from protected SOM, and at the same time release C to the available SOM pool (C25 and C26). The mining algorithm is based on Baskaran et al. (2017).

As the mycorrhizal pools are assumed to have constant C:N ratios, a part of the acquired N is used to fulfill the stoichiometric constraint. Any additional acquired N is leaving the soil system as N supply to the plant. The prescribed C supply from CLM is zero during the winter months, so to ensure that the mycorrhizal fungi do not provide "free" N to the plant during this time, we introduce a scaling factor;

$$r_{myc} = \frac{I_{veg,myc}(t)}{max(I_{veg,myc})} \tag{10}$$

Here, $I_{veg,myc}(t)$ (gCm$^{-2}$h$^{-1}$), is the time varying C supply from vegetation (prescribed from CLM), and $max(I_{veg,myc})$ is the maximum value of $I_{veg,myc}$ in the current year. This scaling factor means that the mycorrhizal fungi are most effective when they receive most energy in the form of C. Since $I_{veg,myc}(t)$ is prescribed in the current model version, the input flux will not respond to changes is soil N availability.

     Constant mortality rates determine the transfer from mycorrhizal fungi to the SOM pools (C19-C24 and N19-N24).

### 2.1.3   Inorganic N processes

Inorganic N is divided between nitrate and ammonium dissolved in soil water ($N_{NO3}$ and $N_{NH4,sol}$), and ammonium sorbed to soil particles ($N_{NH4,sorb}$). Reactive nitrogen from atmospheric deposition enters $N_{NH4,sol}$ (N32) where it can undergo nitrification to $N_{NO3}$ (N34) or become sorbed to particles (N35). $N_{NO3}$ is exposed to leaching and runoff based on CLM algorithms (N31). Both dissolved pools, $N_{NH4,sol}$ and $N_{NO3}$, can be taken up by mycorrhizal fungi (N27, N28) or directly by

plants (N33). Since the model is not coupled to above-ground vegetation, direct plant uptake is a constant loss rate. Within a time step (1 hour) the different processes affecting inorganic N is calculated in a sequence: 1) Deposition, leaching and runoff 2) nitrification 3) N from decomposition 4) direct uptake by vegetation 5) uptake by mycorrhiza 6) exchange with saprotrophs 7) Langmuir sorption algorithm. The Langmuir sorption algorithm is based on Sieczka and Koda (2016) and described below. The basis for this process is cation exchange, where positively charged ammonium is adsorbed to negatively charged clay

particles. Before pt. 7) the total concentration of ammonium is

$$N_{NH4,tot} = N_{NH4,sorp} + N_{NH4,sol} \tag{11}$$

Using Eq. (11) together with the Langmuir isotherm equation, we find the equilibrium partition between $N_{NH4,sol}$ and $N_{NH4,sorp}$ given the total concentration $N_{NH4,tot}$. The Langmuir isotherm equation is given by

$$N_{NH4,sorp,eq} = \frac{NH4_{sorp,max} \cdot K'_L \cdot N_{NH4,sol,eq}}{1 + K'_L \cdot N_{NH4,sol,eq}} \tag{12}$$



where $K'_L$ is a Langmuir constant related to adsorption energy, and a function of soil water content. $NH4_{sorp,max}$ is the maximum adsorption capacity. We assume that the system moves towards the equilibrium value during the timestep, via the following mechanism, derived from the pseudo-second order kinetic model in Sieczka and Koda (2016):

$$N_{NH4,sorp} =$$


$$
\begin{cases}
N_{NH4,sorp,eq} - \dfrac{1}{\frac{1}{N_{NH4,sorp,eq} - NH4_{sorp,prev.}} + k \cdot \Delta t} & N_{NH4,sorp,eq} > N_{NH4,sorp,prev}, \\[3ex]
N_{NH4,sorp,eq} + \dfrac{1}{\frac{1}{N_{NH4,sorp,prev} - N_{NH4,sorp,eq}} + k \cdot \Delta t} & N_{NH4,sorp,eq} < N_{NH4,sorp,prev}, \\[3ex]
N_{NH4,sorp,prev} & N_{NH4,sorp,eq} = N_{NH4,sorp,prev}
\end{cases}
\tag{13}
$$

Here $k$ is a rate constant and $\Delta t$ is the timestep. The top option corresponds to absorption, the middle option to desorption and the third if equilibrium is already reached. All parameter values are from Sieczka and Koda (2016), converted to appropriate model units (see Table A5).

### 2.1.4 Vertical structure

The discrete vertical layers of the model follow the same structure as CLM with increasing layer thickness with depth (Lawrence et al., 2019). This allows incoming litter and N deposition to be distributed following the same vertical profile as in CLM. We use vertically resolved soil temperature and soil moisture from CLM as input to MIMICS+. We also use drainage and runoff rates from CLM to determine N leaching. Within each time step the fluxes between the pools are calculated and applied first, then vertical transport is calculated and applied. This transport is calculated as a simple diffusion equation between adjacent layers (Soetaert and Herman, 2009), using a constant diffusion coefficient from Koven et al. (2013).

## 2.2 Soil profile database

For comparison, a forest soil database collected in connection with the ICP forest monitoring program level 1 sites was used (Lorenz, 1995). These data have been further analyzed by Strand et al. (2016), and provide a unique source of information about boreal soil conditions. A total of 1040 soil profiles were described, sampled and analyzed between 1988 and 1992 (Esser and Nyborg, 1992). Soil profile descriptions were done according to standardized procedures (Sveistrup, 1984) and classified according to the Canadian System of Soil Classification (CSSC). Relevant information from the database includes C and N stocks, Mean Annual Temperature (MAT) and Mean Annual Precipitation (MAP). Specifically, the database contains C content down to 30 cm, 50 cm and 100 cm, making it possible to compare vertically modeled C stocks to observations in these depth intervals. The dataset also contains separate measurements of C and N in the organic LFH (Litter, Fermented, Humic) layer and mineral soil. The organic layer consists of more or less decomposed litter, and although not directly comparable to modeled litter and SOM pools, the C:N ratio in organic vs. mineral soil is still a useful quantity for model evaluation purposes. A more detailed description of the database is given in Strand et al. (2016). Because podzols are the most common soil category in





**Table 1.** MAT and MAP intervals for dividing the sites into climate categories

| Data source | cooler [°C] | "warmer" [°C] | drier [mm · yr$^{-1}$] | wetter [mm · yr$^{-1}$] |
|---|---|---|---|---|
| Observed | (-1.3)–2.5 | 2.6-7.1 | 355–975 | 1009–2510 |
| Model forcing | (-1.8)–3.8 | 3.9-8.1 | 494–1243 | 1244–3606 |

Norwegian forests, we chose to focus on the podzolic sites in the dataset, giving a total of 578 sites. Due to resource limitations,
we chose a representative subset of 50 sites to use for site simulations with CLM and MIMICS+, and the remaining 528 sites
for further comparison with modeled carbon stocks. The 50 sites cover an area from 5° W to 29°W longitude, and from 5° N
to 70° N latitude. The MAT varies from -1.3°C to 7°C, while MAP ranges from 356 to 2510 mm · y$^{-1}$.

## 2.3 Simulation setup

For the subset of 50 sites, we performed single-site simulations using CLM5.1 in BGC (biogeochemistry) mode. Data from
these simulations were used both to force MIMICS+, and to compare the C and N stocks as calculated by the standard de-
composition model in CLM. The observations were performed during the years 1988-1992, so we ran the models up to and
including 1992, and averaged model values over the five years. Unless otherwise stated, these averages is what is used for the
comparisons. The three datasets each containing data from 50 sites will be referred to as OBS (observations from database),
CLM (CLM simulations) and MIMICS+ (MIMICS+ simulations with CLM forcing). An overview of yearly mean input of
carbon and nitrogen is shown in Fig. B1.

For the CLM simulations, a single site configuration with 100 % natural vegetation was used together with atmospheric forc-
ing from the Global Soil Wetness Project forcing data set (GSWP3, https://hydro.iis.u-tokyo.ac.jp/GSWP3/). This is the default
atmospheric forcing for CLM and provides 3-hourly data with 0.5° resolution. Following CLM spin-up protocol (Lawrence
et al., 2019), all sites were spun up for 500 years in "accelerated-decomposition" mode followed by 700 years of "regular
spinup" by recycling atmospheric forcing for 1901–1930. For the period 1850–1900 the atmospheric forcing cycles the years
1901–1920, then historical forcing was used until the end of the simulation.

As with the CLM simulations, MIMICS+ needs to be spun up to equilibrium before running a historical period. The spinup
was performed from arbitrary initial concentrations by recycling monthly averages of soil temperature and moisture, N de-
position, litter and C input from the CLM history files for the years 1850–1869 (which is using atmospheric forcing from
1901–1920) for 1000 years.

### 2.3.1 Climate gradient profiles

To examine how well the models capture variation with temperature, the three datasets (OBS, MIMICS+, CLM) were sorted
by increasing MAT. The first half (N=25) was labeled "cooler", while the second half (N=25) was labeled "warmer". To capture
variation in moisture, the sites were sorted by MAP in the same manner, with the first half labeled "drier" and the second half
labeled "wetter". Because the MAP and MAT data from the observations and the model forcing differs, a few sites ended up in





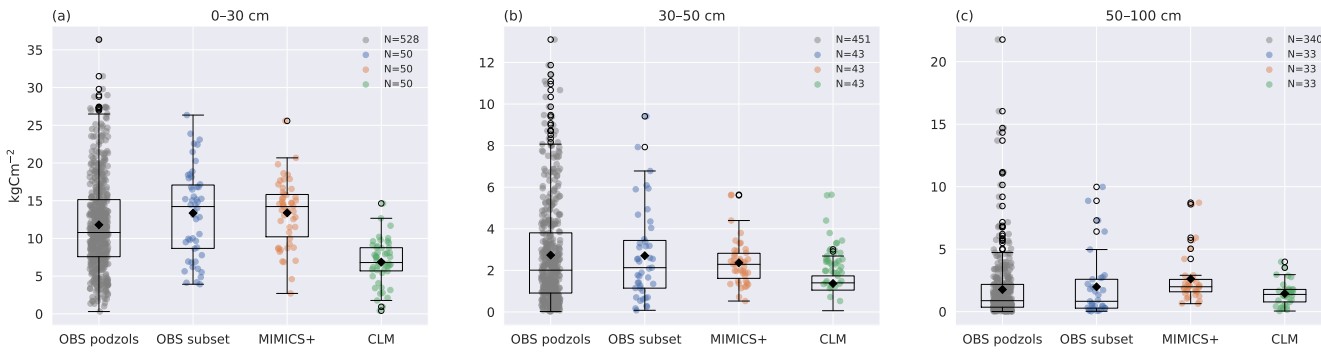

**Figure 2.** Box plots of carbon stocks in (a) 0–30 cm, (b) 30–50 cm, (c) 50–100 cm soil depths for; all observed podzols except the 50 modeled ones (left), the 50 modeled sites (center-left) from Strand et al. (2016), simulated with MIMICS+ (center-right) and with CLM (right). The line in each box is the median, while the diamonds mark the mean values. The box upper and lower edges are the 75th and 25th percentiles, respectively. The whiskers extend from the box by 1.5 times the inter-quartile range. Note the different scales on the y-axes. As not all observed soil profiles are reaching a depth of 30–50 cm or 50–100 cm, these sites are omitted in all boxplots for these depths, hence N=43 for (b) and N=33 for (c).

different categories depending on whether they were sorted by the observed or forcing climate data. The concentration means did not change significantly when using the different datasets, and it was therefore decided to use the sorting by observed climate data for observations and the sorting by forcing climate data for the model simulations. The MAT and MAP intervals for each category are given in Table 1. For some sites the observed soil depth was shallower than 50 cm or 100 cm. These sites

were removed from the distribution box plots for these depth intervals for both the model and observation datasets.

### 2.3.2 N enrichment experiment

To investigate the MIMICS+ modeled response to N enrichment we performed an idealized N addition experiment. Starting from spun-up conditions, we ran two parallel simulations for all 50 sites for 30 years; one "control", using N deposition from the CLM runs, and one "treatment", with an extra amount of 15 $\mathrm{gNm^{-2}}$ deposited. This is a common amount used in

forest fertilization (Högberg et al., 2017). The additional nitrogen was added equally in each time step throughout the second simulation year to give a total of 15 $\mathrm{gNm^{-2}}$. We used these simulations to investigate the temporal response ratios (RR = treatment:control) for different processes and components in the model.

## 3 Results

### 3.1 Carbon and Nitrogen Stocks

Observed and modelled soil carbon stocks are shown in Fig. 2. Both models capture the general trend of decreasing C concentration with increasing depth. The modeled mean C stocks of MIMICS+ across the 50 sites are consistent with the observations





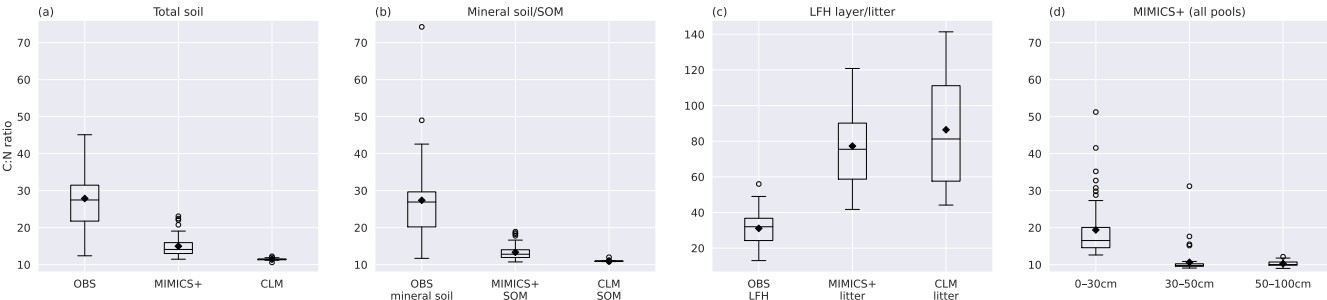

**Figure 3.** Box plots of C:N ratios for observed values from Strand et al. (2016), MIMICS+ and CLM simulations of (a) Total soil, (b) mineral soil (observations) sum of SOM pools (models) (c) the observed forest floor compared to the C:N ratio of simulated litter pools, (d) Total soil at different depths as simulated by MIMICS+. Inorganic N is not considered in any of the plots. The line in each box is the median, while the diamonds mark the mean values. The box upper and lower edges are the 75th and 25th percentiles, respectively. The whiskers extend from the box by 1.5 times the inter-quartile range. N=50 sites.

(OBS subset) at all three depth intervals (not significantly different; $0.28 < p < 0.97$), while the CLM simulations underestimate (significantly different from observations, $p < 0.05$) the stocks in the two upper layers. Due to the heterogeneous nature of real soils and the impact of differences on litter production between the sites, a larger variability in the observations compared to the

simulations is not unexpected. However, site-to-site comparisons with observations are rather poor for both models, although marginally better for MIMICS+ (Appendix, Fig. C1). This is likely explained by subgrid variability in the observations that are not captured by the models. By looking at the collection of sites together, we remove some of the uncertainty related to the variability between the sites and focus on larger patterns in our analyses. There is no significant difference between the two observational subsets, meaning that the 50 sites chosen for the direct model comparison is representative for the broader

region.

Looking at C:N ratios, the overall picture with a higher ratio in the forest floor (observations) and litter pools (models), than in the total soil is captured by both models, again MIMICS+ being closer to the observed values. Both models have significantly lower C:N ratios in the total as well as in the mineral soil (Fig. 3a and b), but MIMICS+ has significantly higher values than CLM ($p < 0.05$). For the litter pools, the pattern is opposite, and the models have significantly higher C:N litter

ratios than observed in the LFH layer. The modeled litter pools are not directly comparable to the LFH layer, but we get an indication of how the modeled C:N ratio compares to the partly decomposed matter. Both models have higher mean values and greater variability than the observations (Figure 3c). This is expected as the observed LFH layer is partly decomposed, and will therefore have lost some C compared to the simulated litter pools which have not yet been affected by the decomposition processes. In addition, the modeled litter pools contain some low-quality (high C:N ratio) CWD, which is not included in the

LFH samples.

The observed total C:N ratio ranges from 12-45 with a mean value of 28, while MIMICS+ and CLM have mean values of 15 and 11, respectively. The range of C:N values from the models are narrower than the observed, with MIMICS+ values



ranging from 11-23 and CLM varies only between 11-12. The large variability among the observations indicates the influence of local conditions on a subgrid-scale. The fact that MIMICS+ has a larger variability than CLM indicates that differences

in soil quality are captured better with the improved modeling framework. Microbial competition for N and a higher fraction of directly plant-derived SOM are factors contributing to this difference between the modeled C:N ratios. Figure 3d shows the MIMICS+ simulated C:N ratios at three different depth layers. As expected, the top layer with more litter has the highest ratio, while in the middle and lowest layers the ratios are significantly lower. For the CLM simulations the C:N ratio is constant around 11 for all three depth intervals. Since we do not have access to observed vertical N stocks, it was not possible to produce

this plot for the observed sites.

## 3.2    Modeled C Pools

In this section we look more in detail into model properties of MIMICS+. With the current model parameterization, the SOM pools contain about 84 % of the total soil C, and 74 % of that is in the protected pools. The litter pools contain most of the remaining C, while 1.3 % is microbial biomass (Fig. 4). The modeled percentage of microbes ranges from 0.3–2 %, and

is in agreement with the 1–3 % microbial biomass C typically reported for soils (Frey, 2019). Figure 4b shows the relative magnitude of each pool within a pool category. Mainly due to the relatively high CWD contribution to the input, the structural pool is the largest litter C pool (12 % of total C, 85 % of total litter C), while metabolic litter consisting of leaf and fine root litter is accounting for ca. 2 % of the total C, and 15 % of total litter. The saprotrophic microbial biomass C is dominating over the mycorrhizal fungi biomass C, and the saprotrophic fungi dominate over saprotrophic bacteria (mean saprotrophic F:B

biomass ratio of 2.0 and above 1 for all sites). This is largely a consequence of the parameter choices in the model, and are further discussed is Sect. 4.

### 3.2.1    Modeled Correlations

Total C (TOTC) is strongly correlated with both MAT and C_input (+0.56 and +0.82, respectively) indicating that higher productivity at warmer sites is an important control on total soil C in the simulations (Fig. 5). The CUE presented in Fig. 5

is calculated as the ratio of total microbial uptake in biomass over the total uptake (including respiration). CUE is positively correlated with available N, pointing to higher microbial efficiencies at sites with higher nutrient content. This is also illustrated by the positive relationship between the percentage of microbial biomass (pct_microbes) and available inorganic N (+0.29 for $N_{NO3}$ and +0.43 for $N_{NH4,sol}$). The negative correlation between CUE and MAT is likely explained by lower quality litter input at warmer sites, as there is a positive relationship between the C:N ratio of the litter input and temperature (+0.46 p < 0.001,

not shown). The lower litter quality causes reduced CUE and hence a negative relationship between temperature and CUE. The strong correlation between production (C_input) and HR (+0.88) indicates that most sites are close to equilibrium. Lower litter quality at high production (and respiration) sites can explain the negative relationship between CUE and HR. There is no significant correlation between CUE and total C.

The fungal:bacterial saprotrophic biomass ratio (FBratio) is negatively correlated to available inorganic N (-0.40 for $N_{NO3}$

and -0.44 for $N_{NH4,sol}$), reflecting the stricter stoichiometrical constrain on bacteria. There is a strong negative correlation




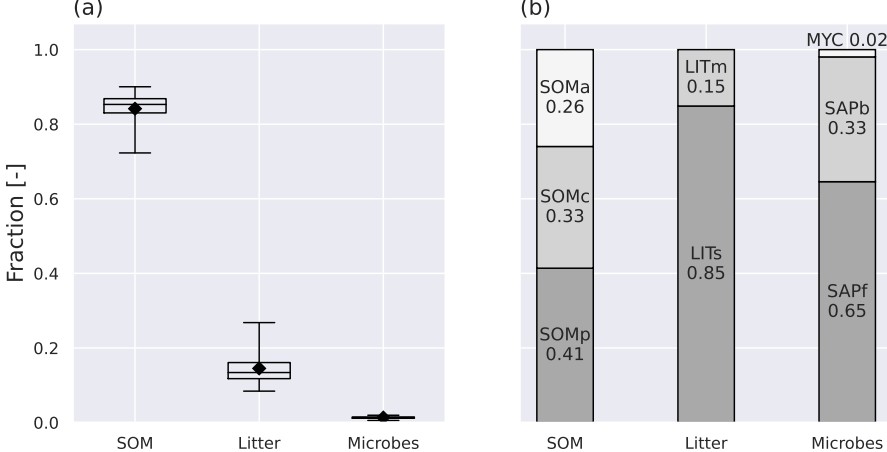

**Figure 4.** Annual mean pool fractions as simulated by MIMICS+. (a) The fractions of total C stored in the main pool categories, soil organic matter (SOM), litter and microbes. The box upper and lower edges are the 75th and 25th percentiles, respectively. The whiskers extend from the box by 1.5 times the inter-quartile range, and N=50 sites. (b) The fraction of C in each pool within each main pool category. MYC covers both EcM and AM, as the AM contribution is so small that it would not be visible on its own.

between the percentage of microbes and the fungal:bacterial ratio (-0.67), reflecting that sites with more available N are more favorable for microbial growth in both pools, but most beneficial for bacteria.

All three inorganic N pools is negatively correlated with MAP (-0.54 for $N_{NO3}$, -0.32 for $N_{NH4,sol}$ and -0.35 for $N_{NH4,sorp}$), $N_{NH4,sorp}$ also with soil water (-0.30). This indicates that the modeled microbes also respond to moisture conditions through the
effects of moisture on inorganic N processes (Leaching, runoff and sorption of $NH_4$) which contribute to making N unavailable, and not only through the modifications of the reverse Michaelis-Menten kinetics.

## 3.3 Climate gradient profiles

In Fig. 6 the 50 sites have been divided into two subsets of 25 sites based on climate categories described in Sect. 2.3.1. Figures 6a–c show lower carbon stocks for colder than for warmer sites for both models and observations for all three depth intervals,
indicating that the models are broadly able to capture the temperature-dependent processes that govern the C storage in soils. In the top 0–30 cm where most of the carbon is stored, MIMICS+ means are close to the observed means. As shown in Fig. 5 the modeled C input is positively correlated with MAT and total soil C, indicating that the difference is mainly caused by differences in litter input. The MIMICS+ simulations show a significant difference between the cold and warm mean (p < 0.05) for all depth intervals, while the cold and warm means from the CLM simulations are not significantly different (0.14 < p <
0.29). This indicates that MIMICS+ temperature dependencies have a larger impact on soil C sequestration than the standard CLM formulation since the C inputs and soil temperatures are the same for the two models.





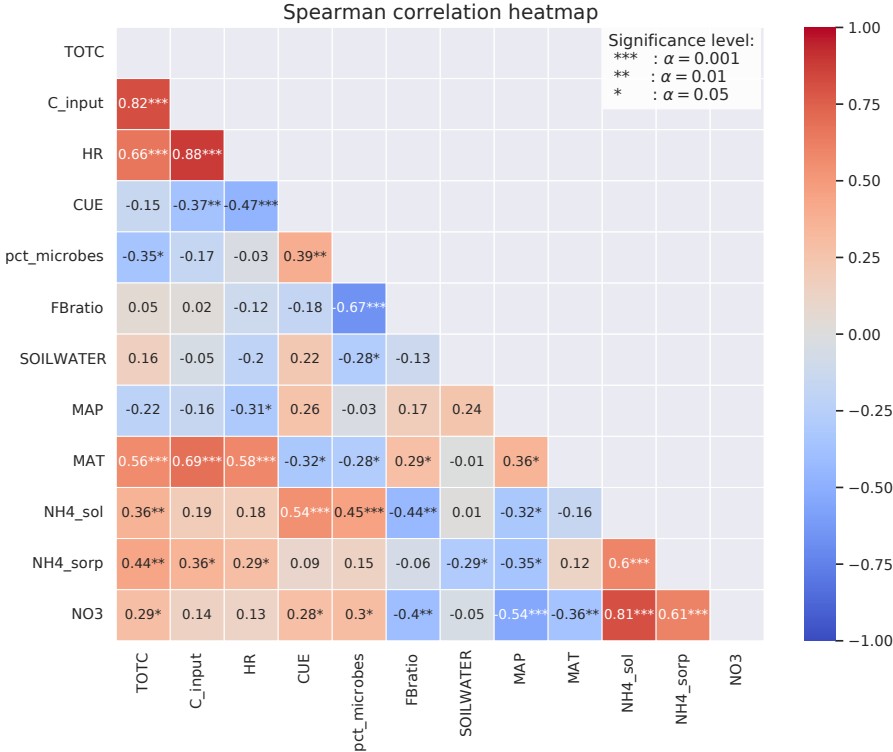

**Figure 5.** Spearman correlation coefficients between different variables calculated from MIMICS+ simulations of the 50 sites. The stars represent significance level of the correlation. Numbers without stars are not significant (p > .05). The color indicate whether the correlation is positive (red) or negative (blue), and the shade indicate the strength of the correlation.

Figures 6d–f show that in the observations, the drier sites have a lower mean C stock than the wetter sites (but not significant). This is opposite of the modeled results; both models show higher mean C content for the drier sites than for the wetter sites. For MIMICS+ this discrepancy is only evident in the top layer (but not significant, p = 0.06), whereas for the lower layers, there are no significant differences between the drier and wetter sites. For the CLM simulations, the pattern is consistent and significant for all three depth intervals (p < 0.05). The influence of moisture on decomposition is represented differently in the two models, which can explain some of the difference between the modeled values. This is further discussed in Sect. 4.3.

### 3.4  N enrichment experiment

The responses to the N enrichment experiment are a result of how the extra reactive N (15 $\mathrm{gNm}^{-2}$ distributed evenly during one year) is distributed between the inorganic nitrogen pools after addition (Fig. 7a–c). All extra N is added to the $N_{\mathrm{NH4,sol}}$ pool, which consequently has the largest response ratio of the three inorganic N pools. Some of this N will move gradually to $N_{\mathrm{NO3}}$ via nitrification or to $N_{NH4,sorp}$ through sorption. While N is lost from $N_{\mathrm{NO3}}$ relatively fast via plant and microbial uptake and leaching, the extra sorbed N serves as a long-term supply of inorganic N, slowly releasing N back to the dissolved





**Figure 6.** Total carbon stocks for cooler/warmer (a–c) and dryer/wetter (d–f) parts of the dataset. Box plots of carbon stocks in the (a), (d) top 30 cm, (b), (e) 30–50 cm, (c), (f) 50–100 cm soil depths for observed profiles from Strand et al. (2016) (left), simulated with MIMICS+ (center) and with CLM (right). In (a–c) the leftmost quartiles represent the coldest 50 % of the dataset, while the rightmost represent the warmest 50 % of the dataset. In (d–f) the leftmost boxes represent the drier 50 % of the total subset, while the rightmost represent the wetter 50 %. The line in each box is the median, while the diamonds mark the mean values. The diamond color represent the climate category; yellow: drier, turquoise: wetter, blue: cooler, red: warmer. The box upper and lower edges are the 75th and 25th percentiles, respectively. The whiskers extend from the box by 1.5 times the inter-quartile range.



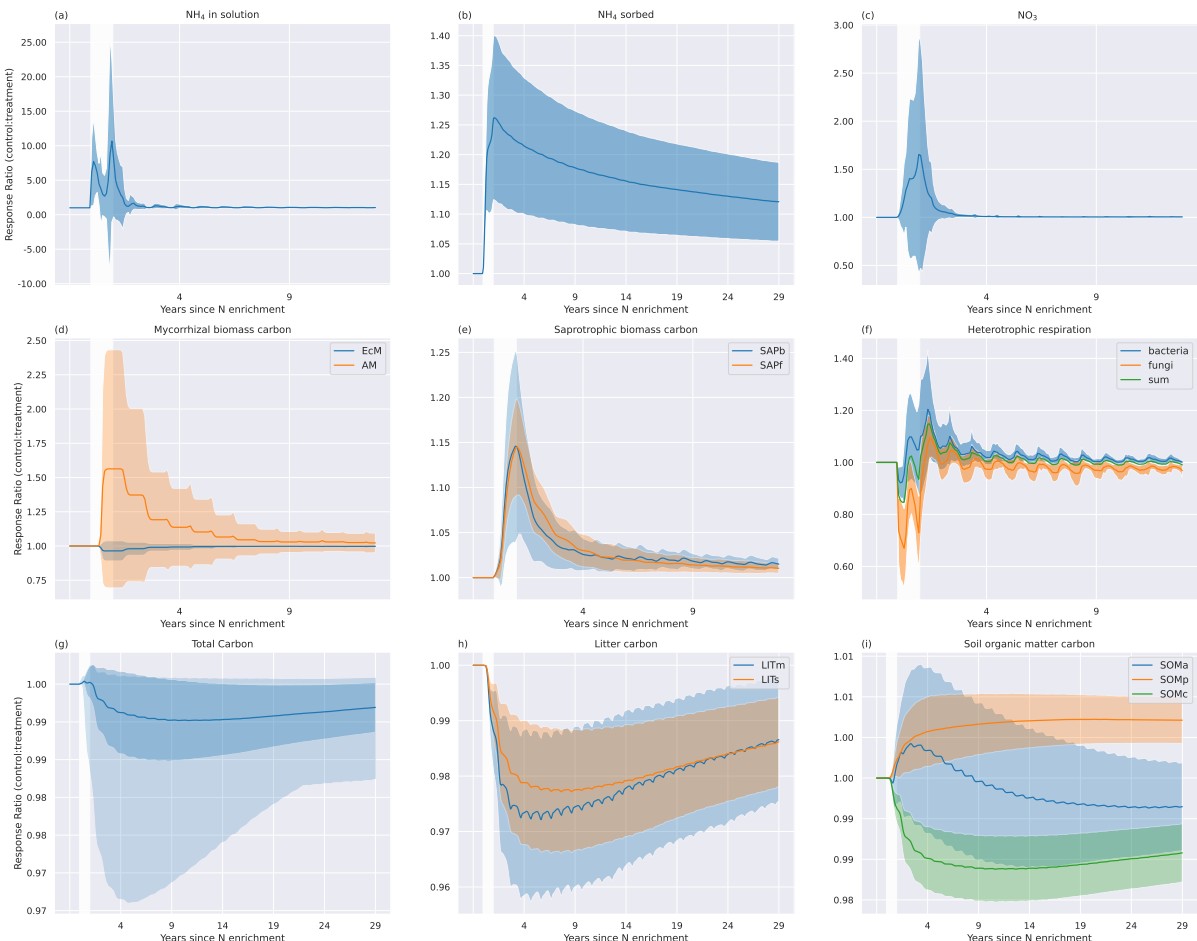

**Figure 7.** Temporal mean (N=50 sites) response ratios (treatment:control) to experimental N enrichment for (a) $N_{NH4,sol}$, (b) $N_{NH4,sorp}$, (c) $N_{NO3}$, (d) mychorrizal fungi pools, (e) saprotrophic pools, (f) heterotrophic respiration, (g) total soil C (h) Litter pools, (i) soil organic matter pools,. The white area marks the year of N enrichment and the shading indicates the standard deviation. In (g) the lighter shading indicate the total spread of the values.





pool. This sustains the higher CUE of the microbes and leads to increased saprotrophic biomass for the duration of the 30-
year simulation. Although $N_{NH4,sol}$ has the largest relative response to N addition, the change in mass of N is largest in the
$N_{NH4,sorp}$ pool.

Looking at each C pool response separately, we see the largest responses in the microbial pools (Fig. 7d–e), with the highest
maximum response for AM, with over 50 % increase in mean biomass in the year of N enrichment. The extra inorganic N
gives a relatively higher return of investment (ROI, Eq. (9)) for AM, resulting in more C allocated to AM and less to EcM.
The initial large response declines gradually but remains positive throughout the simulation period. Although there is a shift to
more AM, the EcM carbon pool is always larger than the AM pool.

Both saprotrophic C pools respond instantly and positively to the N enrichment, with a maximum increase of about 15 % at
the end of the N addition year for both pools. The increase in saprotrophic biomass is a result of higher CUE made possible
by more available N. After the N enrichment year, the response gradually decreases until it stabilizes at around 1 % after ca. 5
years. The long-term response is marginally higher for bacteria than for fungi.

The initial response in HR (Fig. 7f) is a result of a lower respired fraction, (1-CUE), leading to increased saprotrophic
biomass and thus gradually increased rates of litter decomposition. After the initial negative response in HR in the N enrichment
year, there is a positive response due to the higher decomposition rate. For bacterial HR, the response ratio stabilizes at a low
positive value, while for fungi it stabilizes at a slightly negative value. Combined, the response ratio is close to zero for HR
after approximately four years.

The positive microbial biomass responses result in initial decreases in the substrate pools, LITm, LITs and SOMa. The litter
pools remain lower than the base case throughout the simulation period (30 years), while SOMa is also affected by microbial
turnover and changed fluxes from the protected pools. Most microbial necromass ends up in either the physically protected pool
(SOMp) or the available pool (SOMa), leading to an early positive response in these two pools. For SOMp the response remains
positive, while for SOMa the response becomes negative after about eight years. The chemically protected pool experience a
small negative response because increased microbial biomass increases the rate of the depolymerization process that moves
chemically recalcitrant SOM to the available SOM pool (C11). The responses in SOM and litter pools are weak, and following
one year of N enrichment the mean response of total C is a marginal decrease compared to the control. It is worth noting that
some sites experience markedly larger responses in total C than others (shading in Fig. 7g).

The overall response of the model illustrates that shifts in N availability have consequences for microbial C and N dynamics,
although not necessarily for the total C storage and respiration. It should be noted that in this experiment we did not increase
plant productivity and thus carbon input to the soil, which is expected after N enrichment. The added value of this experiment
is that we isolate the in-soil processes and quantify the effects of added nutrients available to microbes and how this affects the
soil carbon pools.





## 4 Discussion


This study aimed to introduce a microbially explicit soil decomposition model, MIMICS+, designed to represent key soil processes in boreal ecosystems that control carbon and nitrogen processing, but still be general enough to be used within an Earth system model. The model was applied to investigate responses to an N enrichment experiment. The results show that MIMICS+ matches observations reasonably well, and for Norwegian forested podzolic sites the model is performing on par

with or better than the state-of-the-art land model CLM using a traditional decomposition formulation (Fig. 2). The C:N ratios from MIMICS+ are closer to observations than CLM, and the predicted fraction of microbial biomass matches well with values reported in the literature (Fig. 3 and Fig. 4). Several interesting correlations between variables were found from the MIMICS+ simulations (Fig. 5). Both models capture the climatic temperature pattern from the observed soil profiles, although they both struggle to represent the observed pattern in C concentrations emerging from the MAP categories (Fig. 6). The N enrichment

experiment demonstrates the implications of adding the Langmuir algorithm for inorganic N, as the sorbed $NH_4$ works as a long-term supply of N for the microbes. The overall effect of the idealized enrichment experiment on soil C storage and respiration was minor but had interesting effects on the relative distribution of the microbial groups, and shows the need for further investigation into the role of sorption-desorption processes of inorganic N, especially in N-limited areas like boreal forests (Fig. 7).

### 4.1 Carbon and Nitrogen Stocks

Looking at the total distribution for the 50 sites, MIMICS+ is closer than CLM to the observations for the two top layers, while none of the modeled means are significantly different from the observations in the bottom layer. As mentioned, site-to-site comparisons with observations are rather poor for both models, showing that there is still a discrepancy between observed and modeled stocks at small scales. This challenge of local factors was illustrated by Pierson et al. (2022) who used the C-only

version of MIMICS with optimized parameters based on local observations and showed reduced error of C stocks on smaller scales (catchments $< 50 \, \mathrm{km^2}$). Such methods would likely also reduce the errors of MIMICS+ at smaller scales. However, it is important to keep in mind that the intention with MIMICS+ is to develop a module that is simple and fast enough to be used in an ESM to simulate the soil carbon dynamics at a grid cell average scale. When forced with grid cell average input, it is not intended to and should not be expected to accurately describe local variation in soil carbon stocks.

With the MIMICS-CN version (Kyker-Snowman et al., 2020) obtained soil C:N ratios that, although within observed ranges, had much lower maximums than the observed ratios. They suggested increasing the fraction of litter going directly to SOM, as forest soils (compared to agricultural and grassland soils) have been shown to contain a high fraction of C in plant residues (Grandy and Robertson, 2007). Our focus area is forested ecosystems, so we increased the fraction of litter going directly to protected SOM without going through microbial decomposition to 50 % for both structural and metabolic litter. This leads to

a longer lifetime of soil C (stored in protected pools) before it becomes available for microbial decomposition and respiration. The higher directly plant-derived fraction in the SOM pools increases the soil C:N ratio, although it is still lower than observed for total and mineral soil (Fig. 3). A recent study by Angst et al. (2021) indicates that the fraction of directly plant-derived SOM




may be much higher than previously assumed, especially for forested sites and podzols. The high C:N ratios in our observational dataset point in the same direction, suggesting that the direct plant-derived fraction is an important factor to consider when

modeling boreal soils. Our results demonstrate that we get closer to observed C:N ratios with MIMICS+ compared to the CLM formulation, a main reason being the high direct plant-derived fraction. In the CENTURY-based decomposition cascade in CLM, the C:N ratios of the SOM pools are fixed, which gives limited options to account for high C:N ratios, and the implications that may have on soil C dynamics.

### 4.2 Modeled C pools

The division of C between the different pools in MIMICS+ shows that most soil C is in the SOM pools (85 %), whereof 73 % is protected. This again reflects the relatively high fraction of litter going directly to protected SOM, but also the lifetime of C in the protected pools before it is either depolymerized or desorbed into the available SOM pool. Compared to MIMICS-CN we doubled the desorption coefficient (see A5), but this is still one order of magnitude lower than the value used in the C-only version of MIMICS (Wieder et al., 2015). In the above-mentioned studies and the present study, this parameter has been

adjusted to match the observed data. In the model formulation, the desorption coefficient is a function of soil clay content, and more observational studies constraining this parameter as a function of clay content and/or other observable variables would benefit further model development.

Saprotrophic fungi are the dominant microbial group in our simulations. Fungi are assumed to have a higher maximum CUE than bacteria in the model (0.7 vs. 0.4, respectively), and are more efficient at decomposing structural litter than the bacterial

pool (higher $V_{max}$ for decomposition of LITs by SAPf than SAPb). This is based on the assumption that fungal decomposers are more specialized towards recalcitrant substrates, while bacteria thrive on labile, easy-access metabolic litter (Wardle et al., 2004). The fraction of CWD litter provided from CLM is relatively large at these forested sites, giving more substrate that is preferable for fungi. The Norwegian podzols we are looking at are nutrient poor, and fungal dominance is expected under N-limited conditions (Strickland and Rousk, 2010). Figure 5 shows a negative correlation between available inorganic N and

F:B ratio, meaning a higher fraction of bacteria in more nutrient rich conditions, in line with observations. Further, the N enrichment experiment indicates that bacteria have a larger positive response to the added N in the long term, which indicates that the model can capture shifts in microbial communities in response to N conditions.

The modeled saprotrophic microbial biomass C dominates over the mycorrhizal fungi biomass C. This contrasts an observational study on boreal forests that indicate that EcM can account for as much as 47–84 % of fungal biomass (Bååth et al.,

2004). Also, Clemmensen et al. (2013) challenged the traditional view that C sequestration is mainly driven by the decomposition of above-ground litter by saprotrophs with their study that showed a dominance of root-associated fungi in deeper parts of the LFH in boreal forests. Few studies exist to inform models about fungal dominance in boreal systems, so parameters determining mycorrhizal growth and turnover are poorly constrained, and not particularly adjusted for boreal conditions in this model iteration. Carbon supply from plants and turnover time are the main influences on mycorrhizal biomass. A sensitivity

test where the mycorrhizal turnover rates was decreased (from $1 \mathrm{yr}^{-1}$ to $0.5 \mathrm{yr}^{-1}$) gave a slight increase in both mycorrhizal and saprotrophic fungi, and although the saprotrophic dominance was reduced, it remained the dominating fungi in the system. The



C supply to mycorrhizal pools is prescribed directly from CLM output, and the growth of these pools is therefore governed by this input rate. Coupling MIMICS+ to the above-ground vegetation will allow the plant C supply to react to nutrient conditions in the soil, and is a priority in future model development.

### 4.2.1 Modeled correlations

Regarding the findings in this section, one should always keep in mind that correlation does not imply causation, especially in a coupled non-linear system like this model. The analysis should be regarded as a broad investigation into possible relationships within the soil dynamics. Recently, Tao et al. (2023) presented CUE as a strong predictor of SOC globally, and argued for a positive correlation between CUE and soil carbon storage (SOC) based on a combination of global-scale datasets, a microbial-process explicit model, data assimilation, deep learning and meta-analysis. Our analysis showed no significant correlation between microbial CUE and soil carbon storage, but a strong correlation between total carbon and plant litter input. A relatively large fraction of the litter input in MIMICS+ (50 %) initially omits the microbial pathway (affected by CUE) as directly plant-derived organic matter into protected SOM pools, which can affect the relationship between microbial CUE and TOTC. A high fraction of saprotrophic necromass ends up in SOMa (Eq. C13-C18 in Table A3 and parameters in Table A5). This leads to a relatively rapid recycling of the carbon that initially goes through the microbial pathway, which can contribute to a weaker relationship between CUE and carbon storage than if larger fractions of the necromass ended up in the protected SOM pools. However, more microbially derived mass in the protected SOM pools will increase the C:N ratio, bringing modeled values further away from the observed C:N ratios in Strand et al. (2016). Tao et al. (2023) used a process-guided deep learning and data-driven modeling (PRODA) approach to optimize parameters in a microbially explicit model (Allison et al., 2010) using observations. Default model parameters prior to optimization gave a negative relationship between CUE and SOC, illustrating how model estimates relies on parameter choices. Using a similar approach to inform MIMICS+ can lead to more robust parameter values in future model iterations.

In MIMICS+ the availability of inorganic N is highly dependent on soil water processes because both N leaching from $N_{NO3}$ and the Langmuir isotherm algorithm is dependent on soil moisture. This is evident from Fig. 5, where we see a negative correlation between inorganic N pools and moisture-related variables (MAP and SOILWATER). The available inorganic N pools are again positively correlated with the percentage of microbes, giving an indirect dependence of microbes on soil moisture. The total C is only weakly negatively correlated with the percentage of microbes, and has a markedly higher correlation with the incoming C. With higher temperatures, we the model gives a higher turnover rate and thereby more release of soil C to the atmosphere. However, increased temperatures also stimulate plant production, especially in boreal and Arctic regions, which can exceed or offset the effect of higher decomposition rates (Hobbie et al., 2002). The correlation patterns from our simulation indicate that the effect of temperature on plant production dominates the effect of temperature on decomposition rates in the model. Pierson et al. (2022) found that increased temperature sensitivity of the decomposition kinetics compared to the original MIMICS/MIMICS+ parameter values reduced error compared to their observational data, indicating that the temperature sensitivity in MIMICS and MIMICS+ may be too weak. However, the agreement between models and observations in Fig. 6a–c indicates that more plant production is the dominating effect of higher temperatures in Norwegian forests.





## 4.3 Climate gradient profiles

Although simple, dividing sites into different climatic categories serves as an idealized "space-for-time" investigation of climate change responses. Assuming that the climate in boreal forests in general, and Norwegian forests specifically will get warmer and wetter in the future (Hanssen-Bauer et al., 2017), the observations indicate higher soil C content at sites with higher MAP
and MAT. The models indicate higher C content for warmer sites, but lower C content for the wetter sites, especially in the 0–30 cm layer. There is a positive correlation between MAT and MAP, particularly for the observed climate forcing (Fig. C4). When dividing the observed sites into the climate categories, a large fraction end up as either "cold and dry" or "warm and wet". We therefore did a simple "temperature-dependence removal" on the total podzol dataset (N=578) by dividing the sites into narrow temperature intervals of 0.5 °C (supplement, Fig. C5). This did not reveal a clear pattern between the wetter and drier sites,
and it is therefore difficult to disentangle the effects of moisture from the effects of temperature in the observed data. Since the models use soil moisture, not MAP to define parameters, we also analyzed the results using a soil moisture variable from the CLM simulations ($SOILWATER\_10CM$) instead of MAP to discriminate between "drier" and "wetter" sites to investigate any effects on the climatic pattern (supplement, Fig. C2). This showed the same pattern as in Fig. 6d-f (more C in drier soils for the models, and less C in drier soils for observations) for all three distributions, except for the deepest layer, where the
trend shifted for the observations, but not significantly. The CLM simulations show a negative correlation between MAP and total C (-0.63, p<0.001, Fig. C3), while this is not evident for MIMICS+, indicating that it is different factors that determine the pattern from the two models. In MIMICS+, the moisture modifier on decomposition works on the fluxes from substrate to the microbial pools. The modeled microbes are most abundant in the top 0–30 cm, which can explain why we observe a difference between drier and wetter sites only in this layer. In CLM, the moisture modification on decomposition rates works
on every step in the decomposition cascade from litter to SOM pools. Since the SOM pools have more C in deeper layers, it can explain why we see the pattern in all three depth intervals for the CLM simulations. The moisture modifier used in MIMICS+ (see $r_{moist}$, Table A5, and Wieder et al. (2017); Sulman et al. (2014)) is a bell-shaped function of soil moisture, limiting decomposition both in the case of very wet and very dry soil conditions. If the optimal soil moisture conditions according to this function are not representing the optimal soil moisture value of the real soils, this could explain why MIMICS+ predicts
the opposite pattern between the drier and wetter soils. Also, soil moisture can vary with subgrid features like slope and aspect, variations not expected to be captured by CLM. Therefore, discrepancy between actual and modeled soil moisture can also be a contributing factor.

## 4.4 N enrichment experiment

Meta-analyses of observational N enrichment studies show that microbial biomass tends to decrease after enrichment, while
the response in total soil C is relatively modest (Treseder, 2008; Janssens et al., 2010). The small modeled response of total soil C to N enrichment (Fig. 7g) is in line with these observations, but the modeled microbial biomass showed a marginal long-term increase after an initial high response (Fig. 7d-e). Treseder (2008) proposed several mechanisms for N effects on microbial growth (Fig. 1 in her study), some leading to an increase while others leading to a decrease in microbial biomass.





The sites studied in our model simulations are mainly N-limited (N immobilization via mechanism 1 in Sect. 2.1.2), and we see an accumulation of microbial biomass as a direct consequence of the increased N availability, which is one of the mechanisms suggested by Treseder (2008) for an increase in microbial biomass. Mechanisms proposed to reduce microbial biomass in response to N enrichment are decreases in soil pH, decrease in ligninase activity, increase in melanoidins and a decrease in below-ground NPP. In this offline iteration of MIMICS+ we are unable to capture potential decreases in below-ground NPP allocation. Coupling to a vegetation model will enable this possibility, and might affect the modeled response N enrichment. In addition, the model gives a very small mycorrhizal fraction (Sect. 4.2), meaning a reduction in C allocation to mycorrhiza does not contribute much, to the overall response of microbial biomass. Therefore, the representation of mycorrhizal parameters and sources and sinks should be reviewed further, to better match the fraction of mycorrhizal biomass indicated by observations (Bååth et al., 2004; Clemmensen et al., 2013). When dividing results into separate biomes, Treseder's analyses indicate that for boreal forests the response for bacteria is positive (RR = 1.061), while for fungi negative (RR = 0.717) but with a confidence interval covering both positive and negative responses (0.0402–1.434). This points to uncertainties also in observations of responses of N enrichment. To cover more of the possible mechanisms for microbial biomass decline in the model, one or more of the other mechanisms mentioned above could be included.

## 4.5 Limitations and Future Improvements of the MIMICS+ framework

By expanding the MIMICS framework with extra microbial groups and an N cycle, we increase possibilities to capture microbe-microbe interactions and, after coupling also plant-microbe interactions. However, we also introduce additional parameters and a more complex model structure that makes the model more prone to overfitting and equifinality issues. While acknowledging this possible drawback/pitfall, we believe valuable insights can be gathered through a more detailed process representation, especially as new technologies allow measurements suitable to constrain model parameters. Although the model produces results comparable to the observations from Strand et al. (2016), there are still poorly constrained parameters in the model, especially related to mycorrhizal C and N transfer. Recent insights about the mycorrhizal role in soil C dynamics are valuable contributions for future model development (Huang et al., 2022a, b). A more robust parameter optimization procedure like the PRODA approach (Tao et al., 2023) or a Monte Carlo approach (Pierson et al., 2022) will contribute to constraining model parameters. The model should also be evaluated against observations from other ecosystems, which will increase confidence in model structure and parameter choices. This offline version of MIMICS+ does not capture plant-microbe interactions and feedbacks, which is essential to capture terrestrial responses to climate change. Therefore, coupling with a vegetation model is a priority in future model development.

## 5 Conclusions

The soil model MIMICS+ provides a tool for investigating soil C processes and interactions with the N cycle, particularly relevant for boreal areas. Furthermore, the model framework will serve as a valuable soil module in ESMs as it is general enough to work on larger scales. The model produces soil C and N stocks comparable to observed values in Norwegian forest





podzols. The explicit representation of microbial groups enhances performance compared to the traditional CLM, and enables detection of soil dynamics not possible with a conventional model. In particular, the novel representation of sorbed inorganic N can be further developed to examine climate responses in N-limited systems like boreal forests, but also in possible impacts on other ecosystems not limited by N. In this study the MIMICS+ model is decoupled from vegetation, so we cannot directly

detect feedbacks between nutrient availability and plant productivity. Coupling MIMICS+ to a dynamical vegetation model like FATES will further enable investigation of the interplay between soil microbes and changing above-ground vegetation.

*Code availability.* MIMICS+ (v1.0) is written in Fortran90. Figures and analyses was produced with Python and Jupyter notebook. The model code and Jupyter notebook is available online at https://doi.org/10.5281/zenodo.8394839.

**Appendix A: Model Description Details**




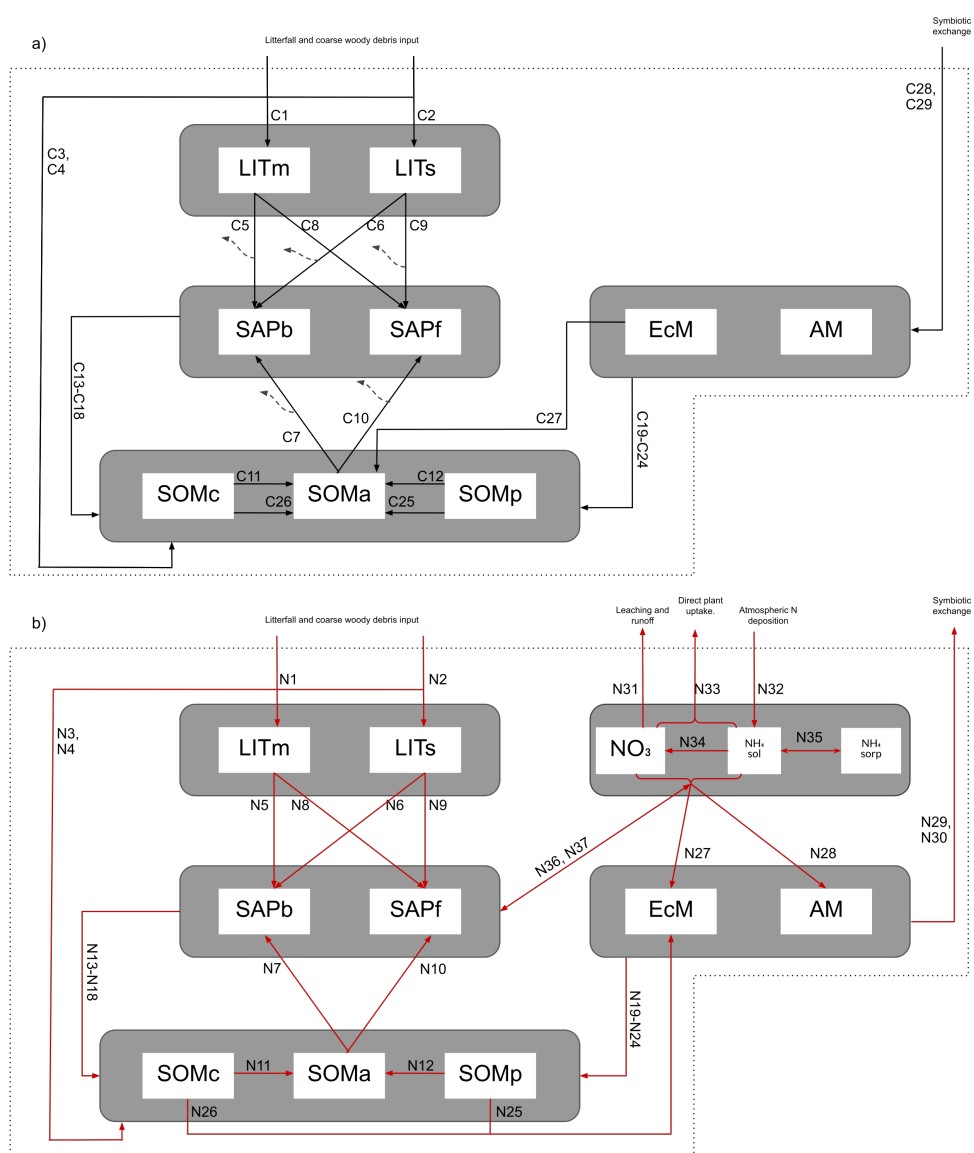

**Figure A1.** Illustration of all carbon (a) and nitrogen (b) pools and fluxes in the system. Expressions for each flux is found with the corresponding numbers in Table A3 and A4.





**Table A1.** Mass balance equations for the carbon pools in the model, calculated for each vertical layer (subscript dropped for readability). $FC_{donor,reciver}$: $gCm^{-3}h^{-1}$. Details about the fluxes are found in Table A3.

| Eqn | Stores | Growth rates | Fluxes |
|---|---|---|---|
| (a) | Metabolic litter | $dC_{LITm}/dt =$ | $FC_{Veg,LITm} - FC_{LITm,SAPb} - FC_{LITm,SAPf}$ |
| (b) | Structural litter | $dC_{LITs}/dt =$ | $FC_{Veg,LITs} - FC_{LITs,SAPb} - FC_{LITs,SAPf}$ |
| (c) | Saprotrophic bacteria | $dC_{SAPb}/dt =$ | $CUE_b \cdot FC^a_{uptake,SAPb} - FC_{SAPb,SOMp} - FC_{SAPb,SOMa} - FC_{SAPb,SOMc}$ |
| (d) | Saprotrophic fungi | $dC_{SAPf}/dt =$ | $CUE_f \cdot FC^a_{uptake,SAPb} - FC_{SAPf,SOMp} - FC_{SAPf,SOMa} - FC_{SAPf,SOMc}$ |
| (e) | Ectomycorrhiza | $dC_{EcM}/dt =$ | $CUE_{EcM} \cdot FC_{Veg,EcM} - FC_{EcM,SOMp} - FC_{EcM,SOMa}$ <br> $- FC_{EcM,SOMc} - FC_{enzEcM,SOMa}$ |
| (f) | Arbuscular mycorrhiza | $dC_{AM}/dt =$ | $CUE_{AM} \cdot FC_{Veg,AM} - FC_{AM,SOMp} - FC_{AM,SOMa} - FC_{AM,SOMc}$ |
| (g) | Phys. protected SOM | $dC_{SOMp}/dt =$ | $FC_{Veg,SOMp} + FC_{SAPb,SOMp} + FC_{SAPf,SOMp}$ <br> $+ FC_{EcM,SOMp} + FC_{AM,SOMp} - FC_{SOMp,SOMa} - FC_{EcMdecompSOMp}$ |
| (h) | Chem. protected SOM | $dC_{SOMc}/dt =$ | $FC_{Veg,SOMc} + FC_{SAPb,SOMc} + FC_{SAPf,SOMc}$ <br> $+ FC_{EcM,SOMc} + FC_{AM,SOMc} - FC_{SOMc,SOMa} - FC_{EcMdecompSOMc}$ |
| (i) | SOM available | $dC_{SOMa}/dt =$ | $FC_{SAPb,SOMa} + FC_{SAPf,SOMa} + FC_{SOMc,SOMa} + FC_{EcM,SOMa} + FC_{enzEcM,SOMa}$ <br> $+ FC_{SOMp,SOMa} + FC_{SOMc,SOMa} + FC_{EcMdecompSOMa}$ <br> $+ FC_{EcMdecompSOMc} + FC_{EcMdecompSOMp} - FC_{SOMa,SAPb} - FC_{SOMa,SAPf}$ |
| | Net Carbon change: | $dC/dt =$ | $FC_{Veg,LITm} + FC_{Veg,LITs} + FC_{Veg,SOMp} + FC_{Veg,SOMc}$ <br> $+ CUE_{EcM} \cdot FC_{Veg,EcM} + CUE_{AM} \cdot FC_{Veg,AM} - (1-CUE_b) \cdot FC_{uptake,SAPb}$ <br> $- (1-CUE_f) \cdot FC^a_{uptake,SAPf}$ |

$^a FC_{uptake,s} = FC_{LITm,s} + FC_{LITs,s} + FC_{SOMa,s}$





**Table A2.** Mass balance equations for the nitrogen pools in the model, calculated for each vertical layer (subscript dropped for readability). $FN_{donor,reciver}$: $gNm^{-3}h^{-1}$. Details about the fluxes are found in Table A4.

| Eqn | Stores | Growth rates | Fluxes |
|---|---|---|---|
| (j) | Metabolic litter | $dN_{LITm}/dt =$ | $FN_{Veg,LITm} - FN_{LITm,SAPb} - FN_{LITm,SAPf}$ |
| (k) | Structural litter | $dN_{LITs}/dt =$ | $FN_{Veg,LITs} - FN_{LITs,SAPb} - FN_{LITs,SAPf}$ |
| (l) | Saprotrophic bacteria | $dN_{SAPb}/dt =$ | $NUE \cdot FN^a_{uptake,SAPb} - FN_{SAPb,SOMp} - FN_{SAPb,SOMa} - FN_{SAPb,SOMc} + FN^{b,c}_{IN,SAPb}$ |
| (m) | Saprotrophic fungi | $dN_{SAPf}/dt =$ | $NUE \cdot FN^a_{uptake,SAPf} - FN_{SAPf,SOMp} - FN_{SAPf,SOMa} - FN_{SAPf,SOMc} + FN^{b,c}_{IN,SAPf}$ |
| (n) | Ectomycorrhiza | $dN_{EcM}/dt =$ | $FN^b_{IN,EcM} + FN_{SOMp,EcM} + FN_{SOMc,EcM}$ |
| | | | $-FN_{EcM,SOMp} - FN_{EcM,SOMa} - FN_{EcM,SOMc} - FN_{EcM,Veg}$ |
| (o) | Arbuscular mycorrhiza | $dN_{AM}/dt =$ | $FN^b_{IN,AM} - FN_{AM,SOMp} - FN_{AM,SOMa} - FN_{AM,SOMc} - FN_{AM,Veg}$ |
| (p) | Phys. protected SOM | $dN_{SOMp}/dt =$ | $FN_{SAPb,SOMp} + FN_{SAPf,SOMp} + FN_{EcM,SOMp} + FN_{AM,SOMp} + FN_{Veg,SOMp}$ |
| | | | $-FN_{SOMp,SOMa} - FN_{SOMp,EcM}$ |
| (q) | Chem. protected SOM | $dN_{SOMc}/dt =$ | $FN_{SAPb,SOMc} + FN_{SAPf,SOMc} + FN_{EcM,SOMc} + FN_{AM,SOMc}$ |
| | | | $+FN_{Veg,SOMp} - FN_{SOMc,SOMa} - FN_{SOMc,EcM}$ |
| (r) | SOM available | $dN_{SOMa}/dt =$ | $FN_{SAPb,SOMa} + FN_{SAPf,SOMa} + FN_{SOMp,SOMa} + FN_{SOMc,SOMa}$ |
| | | | $+FN_{EcM,SOMa} + FN_{AM,SOMa} - FN_{SOMa,SAPb} - FN_{SOMa,SAPf}$ |
| (s) | Ammonium, solved | $dN_{NH4,sol}/dt =$ | $FN_{DEP} + (1 - NUE)(FN_{uptake,SAPb} + FN_{uptake,SAPf})^a -$ |
| | | | $f_{NH4}(FN_{IN,EcM} + FN_{IN,AM} + FN_{IN,Veg}) - f_{NH4}(FN_{IN,SAPb} + FN_{IN,SAPf})$ |
| | | | $-FN_{NH4,NO3} + FN_{sol,sorp}$ |
| (t) | Ammonium, sorbed | $dN_{NH4,sorp}/dt =$ | $-FN_{sol,sorp}$ |
| (u) | Nitrate | $dN_{NO3}/dt =$ | $FN_{NH4,NO3} - FN_{run+leach} -$ |
| | | | $(1 - f_{NH4})(FN_{IN,EcM} + FN_{IN,AM} + FN_{IN,Veg}) - (1 - f_{NH4})(FN_{IN,SAPb} + FN_{IN,SAPf})$ |
| | Net Nitrogen change: | $dN/dt =$ | $FN_{DEP} + FN_{Veg,LITm} + FN_{Veg,LITs} + FN_{Veg,SOMc} + FN_{Veg,SOMp} - FN_{run+leach}$ |
| | | | $-FN_{IN,Veg} - FN_{EcM,Veg} - FN_{AM,Veg}$ |

$^a FN_{uptake,s} = FN_{LITm,s} + FN_{LITs,s} + FN_{SOMa,s}$

$^b FN_{IN,reciever} = FN_{NO3+NH4sol,reciever}$

$^c$ Can be either positive or negative.



**Table A3.** Details about C fluxes in the model. The eq. numbers corresponds to the arrows in Fig. A1a. The letters in the fifth column matches with those given in Table A1. All $FC_{donor,reciver}$ has units $\mathrm{gCm^{-3}h^{-1}}$. Parameters are described in Table A5.

| Eq | Flux Name | Rate Functions | Used in Eqn. | Notes |
|---|---|---|---|---|
| C1 | $FC_{Veg,LITm} =$ | $f_{met} \cdot I_C \cdot (1 - f_{met,SOM})$ | (a) | $I_C$ include litterfall + mortality rates |
| C2 | $FC_{Veg,LITs} =$ | $((1 - f_{met}) \cdot I_C + CWD_C) \cdot (1 - f_{struct,SOM})$ | (b) | |
| C3 | $FC_{Veg,SOMp} =$ | $f_{met} \cdot I_C \cdot f_{met,SOM}$ | (g) | |
| C4 | $FC_{Veg,SOMc} =$ | $((1 - f_{met}) \cdot I_C + CWD_C) \cdot f_{struct,SOM}$ | (h) | |
| C5 | $FC_{LITm,SAPb} =$ | $C_{SAPb} \cdot V_{max1} \frac{C_{LITm}}{K_{m1} + C_{SAPb}}$ | (a)(c) | Reverse MMK |
| C6 | $FC_{LITs,SAPb} =$ | $C_{SAPb} \cdot V_{max2} \frac{C_{LITs}}{K_{m2} + C_{SAPb}}$ | (b)(c) | Reverse MMK |
| C7 | $FC_{SOMa,SAPb} =$ | $C_{SAPb} \cdot V_{max3} \frac{C_{SOMa}}{K_{m3} + C_{SAPb}}$ | (i)(c) | Reverse MMK |
| C8 | $FC_{LITm,SAPf} =$ | $C_{SAPf} \cdot V_{max4} \frac{C_{LITm}}{K_{m4} + C_{SAPf}}$ | (a)(d) | Reverse MMK |
| C9 | $FC_{LITs,SAPf} =$ | $C_{SAPf} \cdot V_{max5} \frac{C_{LITs}}{K_{m5} + C_{SAPf}}$ | (b)(d) | Reverse MMK |
| C10 | $FC_{SOMa,SAPf} =$ | $C_{SAPf} \cdot V_{max6} \frac{C_{SOMa}}{K_{m6} + C_{SAPf}}$ | (i)(d) | Reverse MMK |
| C11 | $FC_{SOMc,SOMa} =$ | $\frac{C_{SAPf} \cdot V_{max2} \cdot C_{SOMc}}{KO \cdot K_{m2} + C_{SAPb}} + \frac{C_{SAPb} \cdot V_{max5} \cdot C_{SOMc}}{KO \cdot K_{m5} + C_{SAPf}}$ | (h)(i) | As in MIMICS |
| C12 | $FC_{SOMp,SOMa} =$ | $C_{SOMp} \cdot k_{desorp}$ | (g)(i) | As in MIMICS |
| C13 | $FC_{SAPb,SOMp} =$ | $C_{SAPb} \cdot k_{SAPb,som} \cdot f_{SAPb,SOMp}$ | (c)(g) | |
| C14 | $FC_{SAPb,SOMc} =$ | $C_{SAPb} \cdot k_{SAPb,som} \cdot f_{SAPb,SOMc}$ | (c)(h) | |
| C15 | $FC_{SAPb,SOMa} =$ | $C_{SAPb} \cdot k_{SAPb,som} \cdot f_{SAPb,SOMa}$ | (c)(i) | |
| C16 | $FC_{SAPf,SOMp} =$ | $C_{SAPf} \cdot k_{SAPf,som} \cdot f_{SAPf,SOMp}$ | (d)(g) | |
| C17 | $FC_{SAPf,SOMc} =$ | $C_{SAPf} \cdot k_{SAPf,som} \cdot f_{SAPf,SOMc}$ | (d)(h) | |
| C18 | $FC_{SAPf,SOMa} =$ | $C_{SAPf} \cdot k_{SAPf,som} \cdot f_{SAPf,SOMa}$ | (d)(i) | |
| C19 | $FC_{EcM,SOMp} =$ | $C_{EcM} \cdot k_{myc,som} \cdot f_{EcM,SOMp}$ | (e)(g) | |
| C20 | $FC_{EcM,SOMc} =$ | $C_{EcM} \cdot k_{myc,som} \cdot f_{EcM,SOMc}$ | (e)(h) | |
| C21 | $FC_{EcM,SOMa} =$ | $C_{EcM} \cdot k_{myc,som} \cdot f_{EcM,SOMa}$ | (e)(i) | |
| C22 | $FC_{AM,SOMp} =$ | $C_{AM} \cdot k_{myc,som} \cdot f_{AM,SOMp}$ | (f)(g) | |
| C23 | $FC_{AM,SOMc} =$ | $C_{AM} \cdot k_{myc,som} \cdot f_{AM,SOMc}$ | (f)(h) | |
| C24 | $FC_{AM,SOMa} =$ | $C_{AM} \cdot k_{myc,som} \cdot f_{AM,SOMa}$ | (f)(i) | |
| C25 | $FC_{EcMdecSOMp} =$ | $K_{MO} \cdot H \cdot C_{EcM} \cdot C_{SOMp} \cdot r_{myc}$ | (g)(i) | (Baskaran et al., 2017) + mod. term |
| C26 | $FC_{EcMdecSOMc} =$ | $K_{MO} \cdot H \cdot C_{EcM} \cdot C_{SOMc} \cdot r_{myc}$ | (h)(i) | (Baskaran et al., 2017) + mod. term |
| C27 | $FC_{enzEcM,SOMa} =$ | $f_{enz} \cdot CUE_{EcM} \cdot FC_{Veg,EcM}$ | (e)(i) | |
| C28 | $FC_{Veg,EcM} =$ | $f_{alloc,EcM} \cdot I_{veg,Myc}$ | (e) | |
| C29 | $FC_{Veg,AM} =$ | $f_{alloc,AM} \cdot I_{veg,Myc}$ | (f) | |





**Table A4.** Details about N fluxes in the model. The eq. numbers corresponds to the arrows in Fig. A1b. The letters in the fifth column matches with those given in Table A2. Parameters are described in Table A5.

| Eq. | Flux Name | Rate functions | Used in eqn | Notes |
|---|---|---|---|---|
| N1 | $FN_{Veg,LITm} =$ | $f_{met} \cdot I_N \cdot (1 - f_{met,SOM})$ | (j) | $I_N$ include litterfall + mortality rates |
| N2 | $FN_{Veg,LITs} =$ | $((1 - f_{met}) \cdot I_N + CWD_N) \cdot (1 - f_{struct,SOM})$ | (k) | |
| N3 | $FN_{Veg,SOMp} =$ | $f_{met} \cdot I_C \cdot f_{met,SOM}$ | (p) | |
| N4 | $FN_{Veg,SOMc} =$ | $((1 - f_{met}) \cdot I_N + CWD_N) \cdot f_{struct,SOM}$ | (q) | |
| N5 | $FN_{LITm,SAPb} =$ | $FC_{LITm,SAPb} \cdot \left( \frac{N_{LITm}}{C_{LITm}} \right)$ | (j)(l) | as in MIMICS |
| N6 | $FN_{LITs,SAPb} =$ | $FC_{LITs,SAPb} \cdot \left( \frac{N_{LITs}}{C_{LITs}} \right)$ | (k)(l) | as in MIMICS |
| N7 | $FN_{SOMa,SAPb} =$ | $FC_{SOMa,SAPb} \cdot \left( \frac{N_{SOMa}}{C_{SOMa}} \right)$ | (r)(l) | as in MIMICS |
| N8 | $FN_{LITm,SAPf} =$ | $FC_{LITm,SAPf} \cdot \left( \frac{N_{LITm}}{C_{LITm}} \right)$ | (j)(m) | as in MIMICS |
| N9 | $FN_{LITs,SAPf} =$ | $FC_{LITs,SAPf} \cdot \left( \frac{N_{LITs}}{C_{LITs}} \right)$ | (k)(m) | as in MIMICS |
| N10 | $FN_{SOMa,SAPf} =$ | $FC_{SOMa,SAPf} \cdot \left( \frac{N_{SOMa}}{C_{SOMa}} \right)$ | (r)(m) | as in MIMICS |
| N11 | $FN_{SOMc,SOMa} =$ | $FC_{SOMc,SOMa} \cdot \left( \frac{N_{SOMc}}{C_{SOMc}} \right)$ | (q)(r) | |
| N12 | $FN_{SOMp,SOMa} =$ | $FC_{SOMp,SOMa} \cdot \left( \frac{N_{SOMp}}{C_{SOMp}} \right)$ | (p)(r) | |
| N13 | $FN_{SAPb,SOMp} =$ | $FC_{SAPb,SOMp} \cdot \left( \frac{N_{SAPb}}{C_{SAPb}} \right)$ | (l)(p) | |
| N14 | $FN_{SAPb,SOMc} =$ | $FC_{SAPb,SOMc} \cdot \left( \frac{N_{SAPb}}{C_{SAPb}} \right)$ | (l)(q) | |
| N15 | $FN_{SAPb,SOMa} =$ | $FC_{SAPb,SOMa} \cdot \left( \frac{N_{SAPb}}{C_{SAPb}} \right)$ | (l)(r) | |
| N16 | $FN_{SAPf,SOMp} =$ | $FC_{SAPf,SOMp} \cdot \left( \frac{N_{SAPf}}{C_{SAPf}} \right)$ | (m)(p) | |
| N17 | $FN_{SAPf,SOMc} =$ | $FC_{SAPf,SOMc} \cdot \left( \frac{N_{SAPf}}{C_{SAPf}} \right)$ | (m)(q) | |
| N18 | $FN_{SAPf,SOMa} =$ | $FC_{SAPf,SOMa} \cdot \left( \frac{N_{SAPf}}{C_{SAPf}} \right)$ | (m)(r) | |
| N19 | $FN_{EcM,SOMp} =$ | $FC_{EcM,SOMp} \cdot \left( \frac{N_{EcM}}{C_{EcM}} \right)$ | (n)(p) | |
| N20 | $FN_{EcM,SOMc} =$ | $FC_{EcM,SOMc} \cdot \left( \frac{N_{EcM}}{C_{EcM}} \right)$ | (n)(q) | |
| N21 | $FN_{EcM,SOMa} =$ | $FC_{EcM,SOMa} \cdot \left( \frac{N_{EcM}}{C_{EcM}} \right)$ | (n)(r) | |
| N22 | $FN_{AM,SOMp} =$ | $FC_{AM,SOMp} \cdot \left( \frac{N_{AM}}{C_{AM}} \right)$ | (o)(p) | |
| N23 | $FN_{AM,SOMc} =$ | $FC_{AM,SOMc} \cdot \left( \frac{N_{AM}}{C_{AM}} \right)$ | (o)(q) | |
| N24 | $FN_{AM,SOMa} =$ | $FC_{AM,SOMa} \cdot \left( \frac{N_{AM}}{C_{AM}} \right)$ | (o)(r) | |
| N25 | $FN_{SOMp,EcM} =$ | $FC_{EcMdecompSOMp} \cdot \left( \frac{N_{SOMp}}{C_{SOMp}} \right)$ | (g)(e) | |
| N26 | $FN_{SOMc,EcM} =$ | $FC_{EcMdecompSOMc} \cdot \left( \frac{N_{SOMc}}{C_{SOMc}} \right)$ | (h)(e) | |
| N27 | $FN_{IN,EcM} =$ | $V_{max,myc} \cdot N_{IN} \cdot \left( \frac{C_{EcM}}{(C_{EcM} + K_{m,myc}/H)} \right) \cdot r_{myc}$ | (s)(u)(n) | Baskaran et al. (2017)+ mod. term, |

– Continued on next page





**Table A4** – Continued from previous page

| Eq. | Flux Name | Rate functions | Used in eqn | Notes |
|---|---|---|---|---|
| | | | | $IN = N_{NO3} + N_{NH4,sol}$ |
| N28 | $FN_{IN,AM} =$ | $V_{max,myc} \cdot N_{IN} \cdot \left( \frac{C_{AM}}{(C_{AM}+K_{m,myc}/H)} \right) \cdot r_{myc}$ | (s)(u)(o) | Baskaran et al. (2017)+ mod. term |
| N29 | $FN_{EcM,Veg} =$ | $(FN_{IN,EcM} + FN_{SOMc,EcM} + FN_{SOMp,EcM})$ | (n) | $IN = N_{NO3} + N_{NH4,sol}$ |
| | | $-CUE_{EcM} \cdot FC_{Veg,EcM} \cdot (1-f_{enz})/CN_{EcM}$ | | |
| | | or lower, if N limited (reduced CUE) | | |
| N30 | $FN_{AM,Veg} =$ | $FN_{IN,AM} - CUE_{AM} \cdot FC_{Veg,AM}/CN_{AM}$ | (o) | $IN = N_{NO3} + N_{NH4,sol}$ |
| | | or lower, if N limited (reduced CUE) | | |
| N31 | $FN_{run+leach} =$ | $N_{NO3} \cdot \left( \frac{QDRAI}{H_2O_{tot}} + \frac{QRUNOFF}{H2O_{top5cm}} \right)$ | (u) | See CTSM doc. 2.22.6 |
| N32 | $FN_{DEP} =$ | $NDEP\_TO\_SMINN \cdot NDEP\_PROF$ | (s) | |
| N33 | $FN_{IN,Veg} =$ | $N_{IN} \cdot k_{uptake}$ | (s)(u) | $IN = N_{NO3} + N_{NH4,sol}$ |
| N34 | $FN_{NH4,NO3} =$ | $NH4 \cdot k_{nitr}$ or zero if temp. is below freezing | (s)(u) | based on CTSM doc. chapter 2.22.5 |
| N35 | $FN_{sol,sorp} =$ | | | |
| N36 | $FN_{IN,SAPb} =$ | $(1-NUE) \cdot U_{Nb} - CUE_b \cdot U_{Cb}/CN_b$ | (l)(s)(u) | $IN = N_{NO3} + N_{NH4,sol}$ |
| | or = | $f_b \cdot N_{for\_sap}$ if limited N | | |
| N37 | $FN_{IN,SAPf} =$ | $(1-NUE) \cdot U_{Nf} - CUE_f \cdot U_{Cf}/CN_f$ | (m)(s)(u) | |
| | or = | $(1-f_b) \cdot N_{for\_sap}$ if limited N | | |



**Table A5.** Description of parameters and other relevant sizes used in the model.

| Parameter | Description | Expression/Value | Units | Notes |
|---|---|---|---|---|
| $f_{met}$ | Met. frac. of plant litter | $0.75 \cdot (0.85 - 0.013 \cdot min(40, lignin:N))$ | - | Wieder et al. (2015) |
| $f_{clay}$ | Clay fraction in soil | 0.0-1.0 | - | |
| $T$ | Soil temperature | - | °C | Vary with season and depth |
| Michaelis Menten kinetics param. for SAP: Wieder et al. (2015), German et al. (2012) | | | | |
| $V_{max}$ | Max reaction velocity | $exp(V_{slope} \cdot T + V_{int}) \cdot a_V \cdot V_{mod} \cdot r_{moist}$ | mg(mg)$^{-1}$h$^{-1}$ | |
| $K_m$ | Half saturation constant | $exp(K_{slope} \cdot T + K_{int}) \cdot a_K \cdot K_{mod}$ | mgCcm$^{-3}$ | |
| $K_{slope}$ | Regression coefficient | LIT: 0.017, SOMa: 0.027 | ln(mgCcm$^{-3}$)°C$^{-1}$ | For all 6 fluxes |
| $V_{slope}$ | Regression coefficient | 0.063 | ln(mg(mg)$^{-1}$h$^{-1}$)°C$^{-1}$ | For all 6 fluxes |
| $K_{int}$ | Regression intercept | 3.19 | ln(mgCcm$^{-3}$) | Directly Wieder et al. (2015) |
| $V_{int}$ | Regression intercept | 5.47 | ln(mg(mg)$^{-1}$h$^{-1}$) | Directly Wieder et al. (2015) |
| $a_V$ | Tuning coefficient | $1.25 \cdot 10^{-8}$ | - | |
| $P$ | Phys. protection scalar used in $K_{mod}$ | $1/(2.0 \cdot exp(-2\sqrt{f_{CLAY}}))$ | - | Wieder et al. (2015) |
| $a_K \cdot K_{mod}$ | Tuning coefficients | $1.953125, 7.81250, 3.90625 \cdot P,$ $7.8125, 3.90625, 2.604167 \cdot P$ | | As in MIMICS imp. in CLM for LITm, LITs, SOMa |
| $V_{mod}$ | Modifies $V_{max}$ | $10.0, 3.0, 10.0, 3.0, 5.0, 2.0$ | - | for LITm, LITs, SOMa entering SAPb, SAPf |
| $KO$ | Increase Km in eq. C11 | 6 | - | Kyker-Snowman et al. (2020) |
| $k_{myc,som}$ | Turnover rate | $1.14 \cdot 10^{-4}$ | h$^{-1}$ | 1y$^{-1}$ as Sulman et al. (2019) and (Baskaran et al., 2017) |
| $k_{SAPb,som}$ | Turnover rate of SAPb | $5.2 \cdot 10^{-4} \cdot exp(0.3 \cdot f_{met}) \cdot$ $max(p_{mod}, m_{mod})$ | h$^{-1}$ | Wieder et al. (2015) + mod. term |
| $k_{SAPf,som}$ | Turnover rate of SAPf | $2.4 \cdot 10^{-4} \cdot exp(0.1 \cdot f_{met}) \cdot$ $max(p_{mod}, m_{mod})$ | h$^{-1}$ | Wieder et al. (2015) + mod. term |
| $p_{mod}$ scales with root profile, $m_{mod} = 0.1$ is the minimum value of the modifier. $m_{mod}$ is used when $T < 0$ | | | | |
| $k_{desorp}$ | desorption rate | $2 \cdot 10^{-6} \cdot exp(-4.5 \cdot f_{clay})$ | h$^{-1}$ | Modified from Kyker-Snowman et al. (2020) |

– Continued on next page



**Table A5** – Continued from previous page

| Parameter | Description | Expression/Value | Units | Notes |
|-----------|-------------|------------------|-------|-------|
| $K_{MO}$ | Mycorrhizal decay rate | $3.42 \cdot 10^{-7}$ | $m^2 gC^{-1}hr^{-1}$ | Baskaran et al. (2017) |
| $V_{max,myc}$ | Max. mycorrhizal uptake of inorg N | $2.05 \cdot 10^{-4}$ | $g \cdot g^{-1}h^{-1}$ | Baskaran et al. (2017) for EcM, we also use it for AM |
| $K_{m,myc}$ | Half saturation constant of ectomycorrhizal uptake of inorg N | 0.08 | $gNm^{-2}$ | Baskaran et al. (2017) for EcM, we also use it for AM |
| $CUE_{EcM}$ | Growth efficiency of mycorrhiza | 0-0.5 | - | Sulman et al. (2019) |
| $CUE_{AM}$ | Growth efficiency of mycorrhiza | 0-0.5 | - | Sulman et al. (2019) |
| $CUE_b$ | Growth efficiency of sap. bacteria | 0-0.4 | - | Determined by N availability. |
| $CUE_f$ | Growth efficiency of sap. fungi | 0-0.7 | - | |
| $NUE$ | Nitrogen use efficiency of saprotrophs | 0.8 | - | Mooshammer et al. (2014a) |
| $r_{moist}$ | Moisture function: $$max\left(0.05, P \cdot \left(\frac{\Theta_{liq}}{\Theta_{sat}}\right)^3 \cdot \left(1 - \frac{\Theta_{liq}}{\Theta_{sat}} - \frac{\Theta_{frozen}}{\Theta_{sat}}\right)^{2.5}\right)$$ | | - | Wieder et al. (2017), Sulman et al. (2014) |
| $r_{myc}$ | Mycorrhizal modifier | 0-1 | - | |
| $f_{SAPb,SOMp}$ | Frac. necromass into SOMp | $0.3 \cdot exp(1.3 \cdot f_{clay})$ | - | |
| $f_{SAPb,SOMc}$ | Frac. necromass into SOMc | $0.1 \cdot exp(-3 \cdot f_{met})$ | - | |
| $f_{SAPb,SOMa}$ | Frac. necromass into SOMa: $$1 - (f_{SAPb,SOMp} + f_{SAPb,SOMc})$$ | | - | |
| $f_{SAPf,SOMp}$ | Frac. necromass into SOMp | $0.2 \cdot exp(0.8 \cdot f_{clay})$ | - | |
| $f_{SAPf,SOMc}$ | Frac. necromass into SOMc | $0.3 \cdot exp(-3 \cdot f_{met})$ | - | |
| $f_{SAPf,SOMa}$ | Frac. necromass into SOMa: $$1 - (f_{SAPf,SOMp} + f_{SAPf,SOMc})$$ | | - | |
| $f_{EcM,SOMp}$ | Frac. necromass into SOMp | 0.4 | - | Assumed |
| $f_{EcM,SOMc}$ | Frac. necromass into SOMc | 0.2 | - | Assumed |
| $f_{EcM,SOMa}$ | Frac. necromass into SOMa | 0.4 | - | Assumed |
| $f_{AM,SOMp}$ | Frac. necromass into SOMp | 0.3 | - | Assumed |
| $f_{AM,SOMc}$ | Frac. necromass into SOMc | 0.4 | - | Assumed |
| $f_{AM,SOMa}$ | Frac. necromass into SOMa | 0.3 | - | Assumed |
| $f_{enz}$ | Frac. of EcM C uptake used for enzyme prod. | 0.10 | - | Assumed |
| $f_{use}$ | Frac. C released by mining taken up by EcM. | 0.10 | - | Assumed |
| $f_{alloc,i}$ | Frac. of C from plant alloc. to myc. $i$ | 0-1 | - | See Sect. 2.1.2 |
| $f_{met,SOM}$ | Frac. of met. litter prod. going directly to SOMp | 0.5 | - | |
| $f_{struct,SOM}$ | Frac. of struct. litter prod. going directly to SOMc | 0.5 | - | |
| $H$ | Soil depth (column height) | | m | Depth of BGC active layers in CLM |






**Table A5** – Continued from previous page

| Parameter | Description | Expression/Value | Units | Notes |
|---|---|---|---|---|
| $D$ | Diffusion coefficient | $1.14 \cdot 10^{-8}$ | $m^2 hr^{-1}$ | Koven et al. (2013): |
| | | | | $1 cm^2 yr^{-1}$ |
| | | | | 1/3 of this value for $N_{NH4,sorp}$ |
| $CN_b$ | Optimal CN ratio for bacteria | 5 | - | Mouginot et al. (2014) |
| $CN_f$ | Optimal CN ratio for sap. fungi | 8 | - | Mouginot et al. (2014) |
| $CN_m$ | Optimal CN ratio for myc. fungi | 20 | - | Baskaran et al. (2017), Wallander et al. (2003) |
| $BD_{soil}$ | Soil Bulk density | $1.6 \cdot 10^6$ | $g \cdot m^{-3}$ | Sieczka and Koda (2016) |
| $NH4_{sorp,max}$ | Max. adsorption capacity | $0.09 \cdot BD_{soil} \cdot 10^{-3}$=144 | $gNH4 \cdot m^{-3}$ | converted from Sieczka and Koda (2016) |
| $K_L^{'}$ | Modified Langmuir constant | $0.4 \cdot soil\_water\_frac^{-1}$ | $m^3 \cdot gNH4^{-1}$ | converted from Sieczka and Koda (2016) |
| $k$ | Rate constant ammonium sorption | $0.0167 \cdot 60 \cdot 10^3 \cdot BD_{soil}^{-1}$ | $m^3 g^{-1} hr^{-1}$ | converted from Sieczka and Koda (2016) |



**Table A6.** CLM variables used in MIMICS+

| CLM-BGC variable | Units | Long name | Notes |
| --- | --- | --- | --- |
| LEAFC_TO_LITTER | $gCm^{-2}s^{-1}$ | leaf C litterfall | |
| FROOTC_TO_LITTER | $gCm^{-2}s^{-1}$ | fine root C litterfall | |
| CWDC_TO_LITR2C_vr | $gCm^{-3}s^{-1}$ | decomp. of coarse woody debris C to litter 2 C | |
| CWDC_TO_LITR3C_vr | $gCm^{-3}s^{-1}$ | decomp. of coarse woody debris C to litter 3 C | |
| M_LEAFC_TO_LITTER | $gCm^{-2}s^{-1}$ | leaf C mortality | |
| M_FROOTC_TO_LITTER | $gCm^{-2}s^{-1}$ | fine root C mortality | |
| M_LEAFC_STORAGE_TO_LITTER | $gCm^{-2}s^{-1}$ | leaf C storage mortality | Input to met. lit. (LITm) |
| M_LEAFC_XFER_TO_LITTER | $gCm^{-2}s^{-1}$ | leaf C transfer mortality | Input to met. lit. (LITm) |
| M_GRESP_STORAGE_TO_LITTER | $gCm^{-2}s^{-1}$ | growth respiration storage mortality | Input to met. lit. (LITm) |
| M_GRESP_XFER_TO_LITTER | $gCm^{-2}s^{-1}$ | growth respiration transfer mortality | Input to met. lit. (LITm) |
| M_FROOTC_STORAGE_TO_LITTER | $gCm^{-2}s^{-1}$ | fine root C storage mortality | Input to met. lit. (LITm) |
| M_FROOTC_XFER_TO_LITTER | $gCm^{-2}s^{-1}$ | fine root C transfer mortality | Input to met. lit. (LITm) |
| M_LIVECROOTC_XFER_TO_LITTER | $gCm^{-2}s^{-1}$ | live coarse root C transfer mortality | Input to met. lit. (LITm) |
| M_DEADCROOTC_XFER_TO_LITTER | $gCm^{-2}s^{-1}$ | dead coarse root C transfer mortality | Input to met. lit. (LITm) |
| M_LIVECROOTC_STORAGE_TO_LITTER | $gCm^{-2}s^{-1}$ | live coarse root C fire mortality to litter | Input to met. lit. (LITm) |
| M_LIVESTEMC_STORAGE_TO_LITTER | $gCm^{-2}s^{-1}$ | live stem C storage mortality | Input to met. lit. (LITm) |
| M_LIVESTEMC_XFER_TO_LITTER | $gCm^{-2}s^{-1}$ | live stem C transfer mortality | Input to met. lit. (LITm) |
| M_DEADSTEMC_STORAGE_TO_LITTER | $gCm^{-2}s^{-1}$ | dead stem C storage mortality | Input to met. lit. (LITm) |
| M_DEADSTEMC_XFER_TO_LITTER | $gCm^{-2}s^{-1}$ | dead stem C transfer mortality | Input to met. lit. (LITm) |
| LEAFN_TO_LITTER | $gNm^{-2}s^{-1}$ | leaf N litterfall | Partitioned based on $f_{MET}$ |
| FROOTN_TO_LITTER | $gNm^{-2}s^{-1}$ | fine root N litterfall | Partitioned based on $f_{MET}$ |
| CWDN_TO_LITR2N_vr | $gNm^{-3}s^{-1}$ | decomp. of coarse woody debris N to litter 2 C | Input to structural litter (LITs) |
| CWDN_TO_LITR3N_vr | $gNm^{-3}s^{-1}$ | decomp. of coarse woody debris C to litter 3 C | Input to structural litter (LITs) |
| M_LEAFN_TO_LITTER | $gNm^{-2}s^{-1}$ | leaf N mortality | Partitioned based on $f_{MET}$. |
| M_FROOTN_TO_LITTER | $gNm^{-2}s^{-1}$ | fine root N mortality | Partitioned based on $f_{MET}$. |
| M_LEAFN_STORAGE_TO_LITTER | $gNm^{-2}s^{-1}$ | leaf C storage mortality | Input to met. lit. (LITm) |
| M_LEAFN_XFER_TO_LITTER | $gNm^{-2}s^{-1}$ | | Input to met. lit. (LITm) |
| M_FROOTN_STORAGE_TO_LITTER | $gNm^{-2}s^{-1}$ | | Input to met. lit. (LITm) |

– Continued on next page



**Table A6** – Continued from previous page

| CLM-BGC variable | Units | Long name | Notes |
|---|---|---|---|
| M_FROOTN_XFER_TO_LITTER | $\text{gNm}^{-2}\text{s}^{-1}$ | | Input to met. lit. (LITm) |
| M_LIVECROOTN_XFER_TO_LITTER | $\text{gNm}^{-2}\text{s}^{-1}$ | | Input to met. lit. (LITm) |
| M_DEADCROOTN_XFER_TO_LITTER | $\text{gNm}^{-2}\text{s}^{-1}$ | | Input to met. lit. (LITm) |
| M_LIVECROOTN_STORAGE_TO_LITTER | $\text{gNm}^{-2}\text{s}^{-1}$ | | Input to met. lit. (LITm) |
| M_LIVESTEMN_STORAGE_TO_LITTER | $\text{gNm}^{-2}\text{s}^{-1}$ | | Input to met. lit. (LITm) |
| M_LIVESTEMN_XFER_TO_LITTER | $\text{gNm}^{-2}\text{s}^{-1}$ | | Input to met. lit. (LITm) |
| M_DEADSTEMN_STORAGE_TO_LITTER | $\text{gNm}^{-2}\text{s}^{-1}$ | | Input to met. lit. (LITm) |
| M_DEADSTEMN_XFER_TO_LITTER | $\text{gNm}^{-2}\text{s}^{-1}$ | | Input to met. lit. (LITm) |
| M_RETRANSN_TO_LITTER | $\text{gNm}^{-2}\text{s}^{-1}$ | | Input to met. lit. (LITm) |
| NPP_NACTIVE | $\text{gCm}^{-2}\text{s}^{-1}$ | | First subtract NPP_NNONMYC, then partition between EcM and AM based on $f_{alloc,i}$ |
| NPP_NNONMYC | $\text{gCm}^{-2}\text{s}^{-1}$ | Non-mycorrhizal N uptake used C | Subtracted from NPP_NACTIVE |
| NDEP_TO_SMINN | $\text{gNm}^{-2}\text{s}^{-1}$ | atmospheric N deposition to soil mineral N | N deposition to NH4 pool |
| LEAF_PROF | $\text{m}^{-1}$ | profile for litter C and N inputs from leaves | Multiplied with LEAF_TO_LITTER to get rates for each layer |
| FROOT_PROF | $\text{m}^{-1}$ | profile for litter C and N inputs from fine roots | Multiplied with FROOT_TO_LITTER to get rates for each layer |
| CROOT_PROF | $\text{m}^{-1}$ | profile for litter C and N inputs from coarse roots | used for input from mortality |
| STEM_PROF | $\text{m}^{-1}$ | profile for litter C and N inputs from stems | used for input from mortality |
| NDEP_PROF | $\text{m}^{-1}$ | profile for atmospheric N deposition | Multiplied with NDEP_TO_SMINN to get deposition for each layer |
| Environmental variables: | | | |
| TSOI | K | soil temperature | Converted to $^\circ C$ |
| WATSAT | $\text{mm}^3\text{mm}^{-3}$ | saturated soil water content (porosity) | Used for calculating $r_{moist}$ |
| SOILLIQ | $\text{kg} \cdot \text{m}^{-2}$ | soil liquid water | Used for calculating $r_{moist}$ |





**Table A6** – Continued from previous page

| CLM-BGC variable | Units | Long name | Notes |
|---|---|---|---|
| SOILICE | $kg \cdot m^{-2}$ | soil ice water | Used for calculating $r_{moist}$ |
| W_SCALAR | - | Moisture (dryness) inhibition of decomp. | Used in nitrification algorithm |
| T_SCALAR | - | temperature inhibition of decomposition | Used in nitrification algorithm |
| QDRAI | $mm \cdot s^{-1}$ | sub-surface drainage | Used for calculating leaching |
| QOVER | $mm \cdot s^{-1}$ | surface runoff | Used for calculating Runoff |
| nbedrock | - | index of shallowest bedrock layer | for determining how many layers to use in the simulations |
| Read from surface data file: | | | |
| PCT_CLAY | - | percent CLAY | |
| PCT_NAT_PFT | - | percent plant functional type on the nat. veg landunit | |






## Appendix B: Input plot

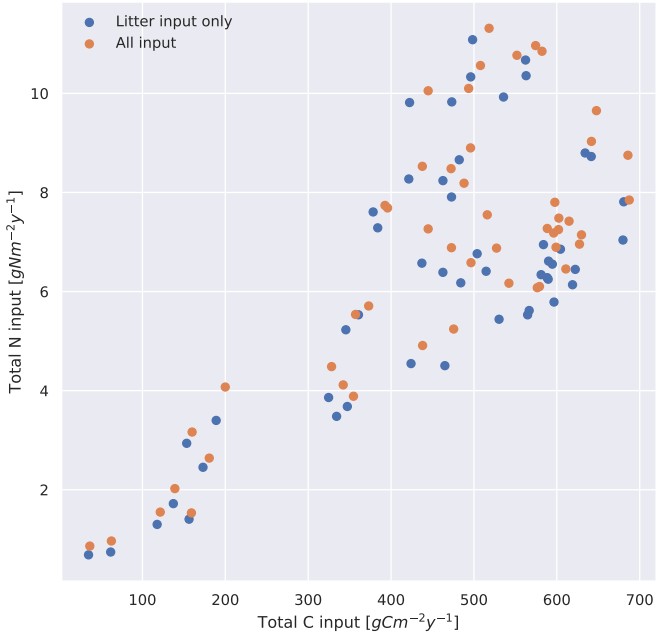

**Figure B1.** Yearly mean input of carbon and nitrogen to MIMICS+ from CLM for each of the 50 site simulations (averaged over 1988-1992). The blue dots show litter input only, while the orange dots also include the C allocated to mycorrhizal pools and N deposition.



## Appendix C: Additional figures

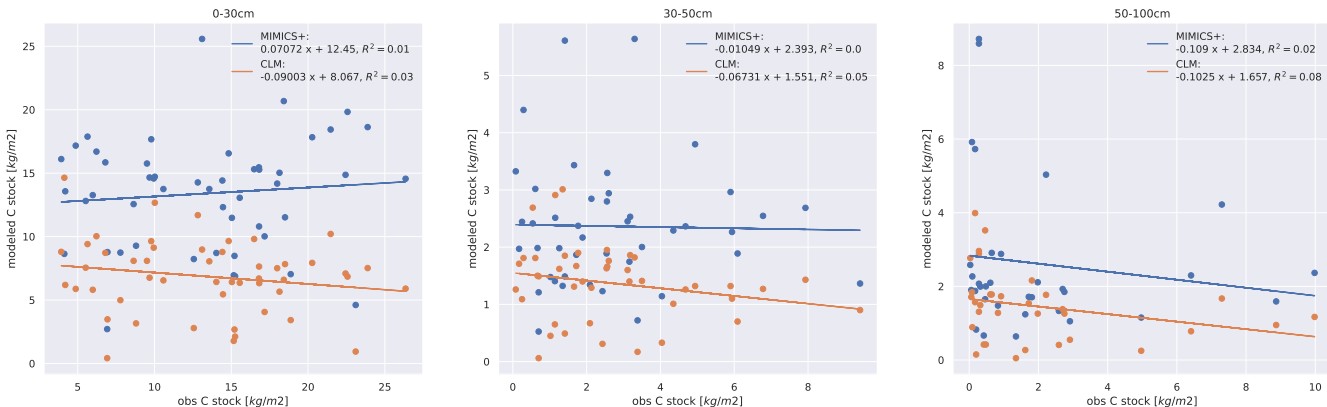

**Figure C1.** Scatter plots of modeled vs. observed C concentrations at a) 0-30 cm depth, b) 30-50 cm depth and c) 50-100 cm depth.

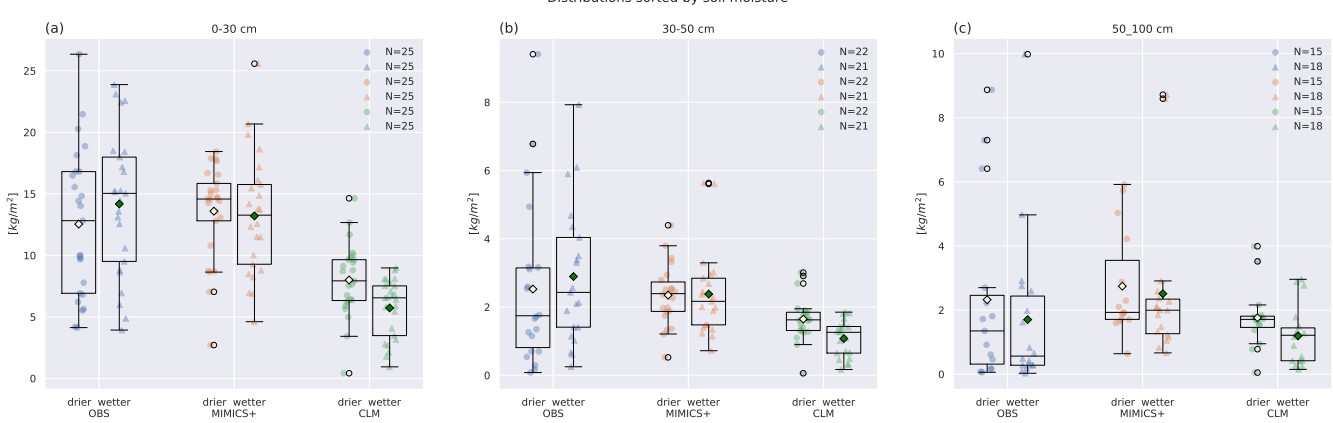

**Figure C2.** Carbon concentrations for dryer/wetter parts of the datasets divided by the CLM variable SOILWATER_10CM. Box plots of carbon concentrations in the (a) 0–30 cm, (b) 30–50 cm, (c) 50–100 cm soil depths for observed profiles from Strand et al. (2016) (left), simulated with MIMICS+ (center) and with CLM (right). The line in each box is the median, while the diamonds mark the mean values. The diamond color correspond to the climate category; yellow: drier, turquoise: wetter. The box upper and lower edges are the 75th and 25th percentiles, respectively. The whiskers extend from the box by 1.5 times the inter-quartile range.





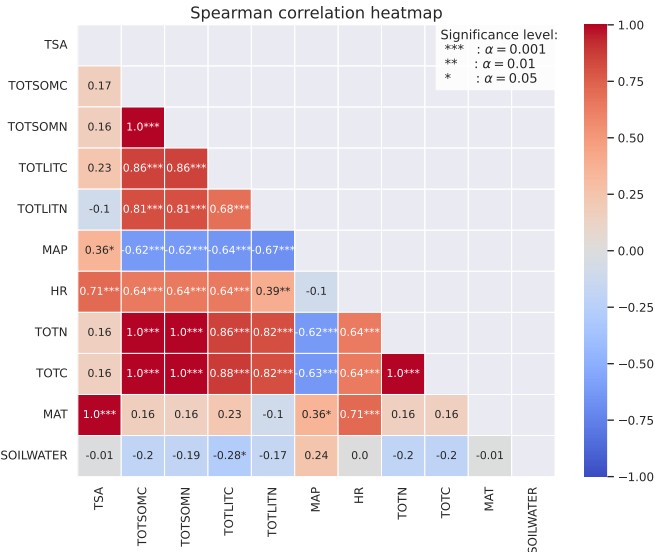

**Figure C3.** Spearman correlation coefficients between different variables calculated from the CLM simulations of the 50 sites. The stars represent significance level of the correlation. Numbers without stars are not significant (p > .05). The color indicate whether the correlation is positive (red) or negative (blue), and the shade indicate the strength of the correlation

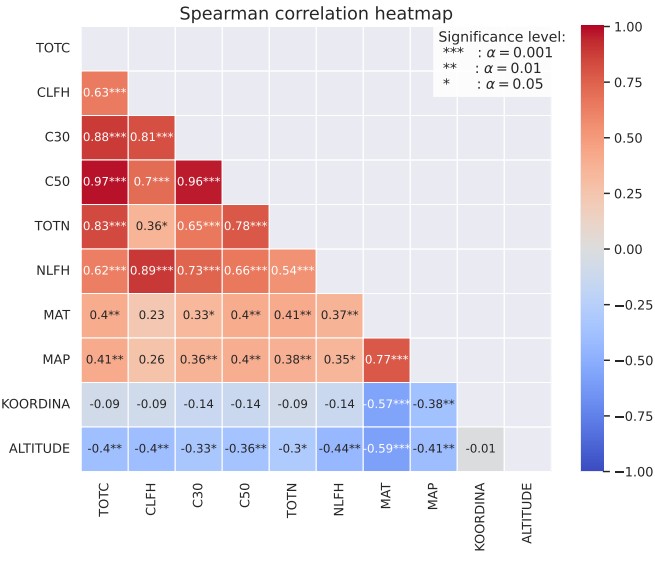

**Figure C4.** Spearman correlation coefficients between different observed variables at 50 sites (Strand et al., 2016). The stars represent significance level of the correlation. Numbers without stars are not significant (p > .05). The color indicate whether the correlation is positive (red) or negative (blue), and the shade indicate the strength of the correlation





**Figure C5.** Each plot contains boxplots of carbon stocks in the depth interval indicated in the top right corner, at sites from the total podzol dataset (N=578), that falls into the MAT interval indicated in the top left corner. The sites in each plot is divided into MAP categories described in main text, Table 1. The line in each box is the median, while the diamonds mark the mean values. The box upper and lower edges are the 75th and 25th percentiles, respectively. The whiskers extend from the box by 1.5 times the inter-quartile range.



*Author contributions.* ERA and TKB developed the model. ERA ran simulations and wrote the manuscript. All authors contributed to the analyses and editing of the manuscript.

*Competing interests.* The authors declare that they have no conflict of interest.

*Acknowledgements.* We are grateful to Line Tau Strand for providing access and information about the soil profile database and to Alexander Eiler, Helge Hellevang, Håvard Kauserud and Rosie Fisher for valuable feedback in the model development process. We are also grateful to Will Wieder and other scientists at the NCAR lab for help and discussion regarding MIMICS and CLM.

*Financial support.* This work has been funded by the University of Oslo and the research council of Norway (RCN) through the research
projects: EMERALD (project no. 294948), FUNDER (project no. 315249) and Green Blue (project no. 287490). The simulations were performed on resources provided by Sigma2 - the National Infrastructure for High Performance Computing and Data Storage in Norway, grant nr. NN2806k/NS2806k. Heleen de Wit was supported by Research Council of Norway (contract nr 160016; Global Change at Northern Latitudes), CatchCaN project (The fate and future of carbon in forests), funded by the Technology Agency of the Czech Republic (TACR) project number TO 01000220.



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
