# Peer review of "Modeling boreal forest soil dynamics with the microbially explicit soil model MIMICS+ (v1.0)"

_EGUsphere, 2023_

## Referee Comment (RC1)

The manuscript presents a model that explicitly represents microbes and couples C-N cycles. The model has the potential to be applied in ESM to work on large scale. The model performs better than the traditional CLM in several aspects and provides new insights regarding to response of soil to N enrichment. However, there are still several aspects could be improved before it can be published.

Line 18: add "," after biosphere

Line 30-33: The sentence is a little confusing, please rephrase it.

Line 33-34: grammar error

Line 45: what did you mean by "direct relationship between N and atmospheric C exchange"?

Line 52: for litter, as well as active, slow, and passive pools of Soil Organic Matter (SOM)

Line 53-62: There are a few microbial models, e.g. MEND, ORCHIMIC that may applied in global scale. What are the advantages and disadvantages of MIMICs compared to them?

Line 241-244: It is weird to classify the data using different standards for observation and model, as this will make the comparison not reasonable.

Line 244: For these sites, they do not have data below 50 or 100cm, or they have low C content below? The followed sentence is not clear, please rephrase.

Line 249: 15 gNm-2/yr

Line 251-252: How did you distinguish the responses of different processes and components? For the N uptake, there is competition from plants. Could you explain how this was modeled in the model? By the way, plant's uptake of N would have large effects on soil C:N, according to ORCHIMIC. MIMIC+ seems to have a much lower C:N in Fig. 3, I am wondering whether there is accumulation of N in the soil in your model?

In Fig 2, the modeled SOC by MIMIC+ seems to comparable to the observation in soil depth 0-30, 30-50, but there is overestimation in 50-100. Could you please explain? I suppose the author should have done some parameter optimization, so what caused this overestimation in deep soil?

Line 288: How is 84% mean? Is it SOC/(SOC+Litter)? What did you mean by "the protected pools"?

Line: 298-299: There should be some pre-conditions for this implication. At global scale, productivities in warm regions are larger than those in cold regions, while SOC are mainly stored in northern high latitude.

Line 365-369: suggest to add few sentences to discuss the responses if increase in plant production (also mean increased N uptake by the plants)

Line 395-408: as I mentioned above, should the lower C:N is because there is some N loss process missing? As you model provided reasonable C stock, but too low C:N, it means you have too much N accumulated in the soil.

The discussion part should be better organized. For example, 4.1 and 4.2 both have C pools. Maybe you can have more clear subsection tiles?

Section 4.5: what is the C:N set for fungi and bacteria? How many microbial groups are in the model? Could the too low C:N is due to the prescribed C:N value? The C:N for microbes should be variable, you may add one sentence to discuss this.

---

## Author Comment (AC1)

***Introduction:***

Please describe the potential mechanisms/processes involved with EcM and AM based on previous research;

Please describe the potential improvements over the models in Sulman et al. (2019) and Baskaran et al. (2017) and distinguish your study with other similar ones.

*We will describe mycorrhizal processes in more detail, including the Gadgil and priming effects that are of key importance when mycorrhizal processes are included in soil models that includes both carbon and nitrogen. Through this we will make a stronger link to work presented by Sulman et al. (2019) and Baskaran et al. (2017). In contrast to Baskaran et al., (2017) MIMICS+ models the N cycle, and is vertically resolved, allowing better representation of microbial competition for nitrogen. Compared to Sulman et al (2019), MIMICS+ presents an alternative structure of pools and fluxes with particular focus on mechanisms that are important in boreal/cold climates.*

**Methods:**

Sensitivity analysis of parameters to modeled soil carbon is routine for model development and therefore is necessary here.

*We agree that sensitivity analysis of parameters (especially poorly constrained ones) should be included. The revised manuscript will include sensitivity analysis for key parameters, such as e.g. CUE, C:N ratios and direct litter-to-SOM fractions.*

Line 103: describe in full C11, C12, N11, N12.

*These fluxes will be described in more detail.*

Lines 162-164: add details about how EcM produces enzymes and the mining algorithm (since it is the import part of this study); is there carbon cost for modeled nitrogen mining?

*These details are described in the detailed figures and tables in the Appendix. The flux C27 is the carbon cost for ECM (from ECM to SOMa). We will add a sentence with references to the appendix in the main text.*

Lines 180-184: do they occur at the same time or sequentially? Please clarify. This is very important for nutrient competition.

*Numerically, they happen in the sequence presented in the manuscript, but since the time step is so short (1h) the order is of limited importance. Our understanding is that the processes in boreal forests are relatively slow, so that the residence times of the reservoirs are much longer than the 1 hour time step. Therefore the inaccuracy implied by the sequential approach is small. We will add a short note about this in the text.*

Lines 195-201: please describe here how the parameters change with depth, in particularly the ones associated with EcM and AM.

*The mycorrhizal parameters like turnover, mining, N uptake etc. are not explicitly dependent on environmental variables (temperature and moisture) and thereby not depth dependent. We will explain this in more detail in the text. During the revision we found that the weighting with respect to the thickness of the soil vertical layers were not entirely correct for the N uptake parameters, and has been changed for the rerun of the model (See our response under the "Results" header).*

Lines 214-216: cannot follow this sentence. Please elaborate.

*The sentence will be rewritten to say "Due to computational resource limitations, we chose a subset of 50 representative sites (out of the 578) for the site simulations with CLM and MIMICS+. The remaining 528 sites was used for further comparison with the modeled carbon stocks." We hope that it is clarifying.*

**Results:**
The poor simulation results in terms of soil carbon (site level one to one), and CN ratio is concerning. Please justify.

*As the model is intended to work on larger spatial scales within an ESM model, a good one-to-one match with specific sites is of less importance than being able to capture larger patterns in temperature and moisture. Norwegian forest soils are also very heterogeneous and local features could have affected measurements in a way that not necessarily represent the general pattern we want to capture in a model. However, we agree that the poor one-to-one match is unsatisfactory. A recently discovered error in CLM in the amount of C allocated to mycorrhizal uptake, thus affecting the forcing of the MIMICS+ model (in terms of how much and in what form carbon was transferred from the vegetation to the soil) might explain at least part of the discrepancy between modeled and observed soil carbon and CN. For the revised manuscript we therefore want to rerun our simulation with the correct (and higher) C allocation to mycorrhiza (with this error in CLM the allocation was too low). The extra carbon is a relatively small fraction of the overall carbon input to the system, but will alter the nutrient competition between microbes and could have an impact on model performance.*

Please show seasonal variation of microbial respiration for model quality judgment. It can be shown in appendix though.

*We agree with the point about seasonal variation of microbial respiration, and will include a figure in the appendix showing this.*

*In the main text please show site-to-site correlation in addition to the boxplot. It is more informative.*

*The site-to-site correlation will be presented in the main text, in addition to the boxplot.*

---

## Author Comment (AC2)

Line 18: add "," after biosphere

Line 30-33: The sentence is a little confusing, please rephrase it.

Line 33-34: grammar error.

*We will rewrite the sentences mentioned above.*

Line 45: what did you mean by "direct relationship between N and atmospheric C exchange"?

*This is referring to the process where microbes might respire a larger fraction of the substrate (thereby releasing more carbon to the atmosphere) when nutrients (nitrogen) is limited in the soil. We will rewrite the sentence to clarify.*

Line 52: for litter, as well as active, slow, and passive pools of Soil Organic Matter (SOM).

*Thanks, will use your suggestion.*

Line 53-62: There are a few microbial models, e.g. MEND, ORCHIMIC that may applied in global scale. What are the advantages and disadvantages of MIMICs compared to them?

*In the revised introduction we will put stronger emphasis on other microbial models, and the differences between existing models and MIMICS+.*

Line 241-244: It is weird to classify the data using different standards for observation and model, as this will make the comparison not reasonable.

*We agree that this is not intuitive, and had extensive discussions about this during preparation of the draft. Our main argument for the choice is that we are investigating specific sensitivities to temperature and precipitation, and therefore split the two datasets based on their local climate forcing, and not geographic location. We will add a figure in the appendix where the sites are classified using geographic location and clarify our choice in the text.*

Line 244: For these sites, they do not have data below 50 or 100cm, or they have low C content below? The followed sentence is not clear, please rephrase.

*The measurements of these soil profiles hit bedrock at a shallower depth than 50/100 cm (so there is no observed data at those depth intervals). We will rephrase to clarify.*

Line 249: 15 gNm-2/yr *Thanks, will fix!*

Line 251-252: How did you distinguish the responses of different processes and components?

*We agree that this sentence is unclear. The way the diagnosis in the model is set up, we distinguish the responses in terms of RR of carbon and nitrogen content of the different pools, and not the processes (fluxes). This will be clarified in the text.*

For the N uptake, there is competition from plants. Could you explain how this was modeled in the model? By the way, plant's uptake of N would have large effects on soil C:N, according to ORCHIMIC. MIMIC+ seems to have a much lower C:N in Fig. 3, I am wondering whether

there is accumulation of N in the soil in your model?

*As you say, plants play a big role for the soil nutrient cycle, both because of direct N uptake, but also because of the C:N ratio of the incoming litter. In the current model version, the plant N uptake is modeled as a 1st order loss from the inorganic N pools. A weakness of the offline model version presented in this study is the limited options to explore plant-microbe interactions, but this is a priority in future work with the model. Another factor to consider is that the C:N ratios in the observations are quite high compared to many other studies, like the ones used for informing the CLM model. A mismatch between C:N ratio of the observed and modeled vegetation could also explain some of the discrepancy. In the revised manuscript we will add more details about how the model handles N uptake by plants, and in the discussion address how this could affect our results compared to other models such as ORCHIMIC.*

In Fig 2, the modeled SOC by MIMIC+ seems to comparable to the observation in soil depth 0-30, 30-50, but there is overestimation in 50-100. Could you please explain? I suppose the author should have done some parameter optimization, so what caused this overestimation in deep soil?

*This could at least partly be explained by the relatively high fraction of litter going directly to the protected pools. At depth, there is little microbial activity, and carbon might accumulate before eventually being lost through respiration. The revised manuscript will include parameter analysis (see authors response under "Methods" to Referee 2). We will tie this to possible causes to the overestimation at deeper soils. Since the model estimate is not significantly higher than the OBS subset ($p < 0.05$), we consider this overestimation to be of limited importance.*

Line 288: How is 84% mean? Is it SOC/(SOC+Litter)? What did you mean by "the protected pools"?

*This means 84% of all carbon in the system, i.e. all nine pools. The "protected pools" refers to the pools SOMp and SOMc (figures and mass balances are found in Appendix A), which is protected from microbial decomposition, physically (SOMp) and (bio)chemically (SOMc) in the same manner as the original MIMICS version presented by Wieder et al. 2015. The sentence will be clarified in the revised manuscript.*

Line: 298-299: There should be some pre-conditions for this implication. At global scale, productivities in warm regions are larger than those in cold regions, while SOC are mainly stored in northern high latitude.

*We will specify this by changing the sentence to "For the focus region (boreal sites in Norway) of this study, total C (TOTC) is strongly correlated with both MAT and C_input (+0.56 and +0.82, respectively) indicating that higher productivity at warmer sites is an important control on total soil C in the simulations (Fig. 5)."*

Line 365-369: suggest to add few sentences to discuss the responses if increase in plant production (also mean increased N uptake by the plants).

*Good point! We will add some discussion about this in the discussion section.*

Line 395-408: as I mentioned above, should the lower C:N is because there is some N loss process missing? As you model provided reasonable C stock, but too low C:N, it means you have too much N accumulated in the soil.

*As mentioned in the response to your previous comment, the discrepancy between observed and modeled C:N ratio can have several causes. For the revised manuscript we will rerun the simulations with corrected C allocation to mycorrhiza (see our response to reviewer 2 under "Results"). We expect this to bring modeled C:N values closer to observed due to increased N competition and loss through mycorrhiza. Generally, we think that parameter choices rather than missing processes in the model could explain these discrepancies. We will address this in connection to the parameter analysis in the revised manuscript.*

The discussion part should be better organized. For example, 4.1 and 4.2 both have C pools. Maybe you can have more clear subsection tiles?

*In this paper we partly perform model-observation comparisons (Sections 3.1 & 4.1), and partly analyze the model result itself (Sections 3.2 and 4.2). We discussed several approaches to the organization of the sections, and found the current structure to be the most suitable for the story(?). To make the structure clearer to the reader, we will add a brief explanation to the beginning of the results section, and also revise the subsection titles.*

Section 4.5: what is the C:N set for fungi and bacteria? How many microbial groups are in the model? Could the too low C:N is due to the prescribed C:N value? The C:N for microbes should be variable, you may add one sentence to discuss this.

*Details about model structure (four microbial pools) and parameters (C:N ratio etc.) are found in Methods and Appendix A. We will add some discussion about the consequences of the low, constant, microbial C:N ratios used in the model, along with references to the text/Appendix where it's needed. We will also include a parameter analysis for different values of microbial C:N which will inform the discussion.*

---

## Author Response (AR1)

Dear editor,

Below follows our responses and manuscript changes to the referee comments. As mentioned in our response to reviewer 2, we decided to rerun the MIMICS+ simulations with a corrected input to EcM after a bug in the mycorrhizal allocation parameters in standard CLM version was discovered (see https://github.com/NCAR/LMWG_dev/issues/8). This led to too low C allocation to mycorrhiza in MIMICS+ in the original simulations. In these reruns, we also corrected an inaccuracy in the mycorrhizal N uptake parameters and adjusted the EcM mining parameter to account for the changes in C input to mycorrhiza (see changes in Table A3 and A4). The material referred to under "Code availability" have been updated accordingly.

With the new simulations the overall conclusions of the study remain largely unchanged, but the new model results required updated versions of  Fig. 2-7, C1 (Previously B1) and S2 (previously C2), in addition to updated numerical values referred to in Results and Discussion. Consequently, this also led to some rewriting of the Results and Discussion sections.

In addition to the changes related to review comments and the model reruns, some minor changes and typo corrections were done, these are listed at the bottom of this document. Figure C2 and C5 has been moved to Supplement material.

The line numbers referred to under "Changes in manuscript" are corresponding to the marked-up manuscript showing the changes.

**Referee 1:**

**Referee comment:**

Line 18: add "," after biosphere

Line 30-33: The sentence is a little confusing, please rephrase it.

Line 33-34: grammar error.

**Author's response:**

*We will rewrite the sentences mentioned above.*

**Changes in manuscript:**

Line 18: Added comma

Line 30-33: Split sentence in two, it now reads "For example, studies show that the kinetics of soil microbes accustomed to cooler climates are more temperature sensitive than microbes in warmer climates (German et al., 2012). Koven et al. (2017) also showed that soil carbon turnover times in cold areas are more sensitive to climatological temperature than in warm areas.

Line 33 (now line 34): added "where"

**Referee comment:**

Line 45: what did you mean by "direct relationship between N and atmospheric C exchange"?

**Author's response:**

*This is referring to the process where microbes might respire a larger fraction of the substrate (thereby releasing more carbon to the atmosphere) when nutrients (nitrogen) is limited in the soil. We will rewrite the sentence to clarify.*

**Changes in manuscript (line 51-52):**

Sentence was rewritten to "This direct relationship between soil N and the C exchange between the atmosphere and soils emphasizes the importance of including microbial C-N relationships in C cycle models."

**Referee comment:**

Line 52: for litter, as well as active, slow, and passive pools of Soil Organic Matter (SOM).

**Author's response:**

*Thanks, will use your suggestion.*

**Changes in manuscript (line 60):**

Sentence now reads: "Traditionally, decomposition processes in models have been represented by first-order kinetics for litter, as well as active, slow, and passive pools of Soil Organic Matter (SOM) (Parton et al., 1988)."

**Referee comment:**

Line 53-62: There are a few microbial models, e.g. MEND, ORCHIMIC that may applied in global scale. What are the advantages and disadvantages of MIMICs compared to them?

**Author's response:**

*In the revised introduction we will put stronger emphasis on other microbial models, and the differences between existing models and MIMICS+.*

**Changes in manuscript:**

We found it more natural to introduce the models directly relevant to MIMICS+ (other versions of MIMICS and the models by Sulman et al. 2019 and Baskaran et al. 2017) in the introduction (line 60-80). References to the models suggested by the reviewer is now added in line 65, and ORCHIMIC is also mentioned in the Discussion section (line 579).

**Referee comment:**

Line 241-244: It is weird to classify the data using different standards for observation and model, as this will make the comparison not reasonable.

**Author's response:**

*We agree that this is not intuitive, and had extensive discussions about this during preparation of the draft. Our main argument for the choice is that we are investigating specific sensitivities to temperature and precipitation, and therefore split the two datasets based on their local climate forcing, and not geographic location. We will add a figure in the appendix where the sites are classified using geographic location and clarify our choice in the text.*

**Changes in manuscript:**

We make a stronger argument for our choice in the Method's section (line 275-280). The figure is added to the supplementary material S1).

**Referee comment:**

Line 244: For these sites, they do not have data below 50 or 100cm, or they have low C content below? The followed sentence is not clear, please rephrase.

**Author's response:**

*The measurements of these soil profiles hit bedrock at a shallower depth than 50/100 cm (so there is no observed data at those depth intervals). We will rephrase to clarify.*

**Changes in manuscript:**

The sentence was rephrased to "For some sites the measured soil depth was shallower than 50 cm or 100 cm. These sites, where the depth to bedrock was less than 50 cm or 100 cm, were removed from both the model and observation datasets before making distribution box plots for these depth intervals." (line 281-283)

**Referee comment:**

Line 249: 15 gNm-2/yr

**Author's response:**

*Thanks, will fix!*

**Changes in manuscript:**

Error was fixed.

**Referee comment:**

Line 251-252: How did you distinguish the responses of different processes and components?

**Author's response:**

*We agree that this sentence is unclear. The way the diagnosis in the model is set up,  we distinguish the responses in terms of RR of carbon and nitrogen content of the different pools, and not the processes (fluxes). This will be clarified in the text.*

**Changes in manuscript:**

The sentence was changed to "We used these simulations to investigate the temporal response ratios (RR =treatment:control) for different C and N pools, as well as for HR." (line 290)

**Referee comment:**

For the N uptake, there is competition from plants. Could you explain how this was modeled in the model? By the way, plant's uptake of N would have large effects on soil C:N, according to ORCHIMIC. MIMIC+ seems to have a much lower C:N in Fig. 3, I am wondering whether there is accumulation of N in the soil in your model?

**Author's response:**

*As you say, plants play a big role for the soil nutrient cycle, both because of direct N uptake, but also because of the C:N ratio of the incoming litter. In the current model version, the plant N uptake is modeled as a 1$^{st}$ order loss from the inorganic N pools. A weakness of the offline model version presented in this study is the limited options to explore plant-microbe interactions, but this is a priority in future work with the model. Another factor to consider is that the C:N ratios in the observations are quite high compared to many other studies, like the ones used for informing the CLM model. A mismatch between C:N ratio of the observed and modeled vegetation could also*

*explain some of the discrepancy. In the revised manuscript we will add more details about how the model handles N uptake by plants, and in the discussion address how this could affect our results compared to other models such as ORCHIMIC.*

**Changes in manuscript:**

The direct plant uptake is described in lines 202-203 (Method). Implications of this simplified approach are discussed in lines 577-590 (Sect. 4.4).

**Referee comment:**

In Fig 2, the modeled SOC by MIMIC+ seems to comparable to the observation in soil depth 0-30, 30-50, but there is overestimation in 50-100. Could you please explain? I suppose the author should have done some parameter optimization, so what caused this overestimation in deep soil?

**Author's response:**

*This could at least partly be explained by the relatively high fraction of litter going directly to the protected pools. At depth, there is little microbial activity, and carbon might accumulate before eventually being lost through respiration. The revised manuscript will include parameter analysis (see authors response under "Methods" to Referee 2). We will tie this to possible causes to the overestimation at deeper soils. Since the model estimate is not significantly higher than the OBS subset (p<0.05), we consider this overestimation to be of limited importance.*

**Changes in manuscript:**

With the rerun simulations (run after the Author's response was formulated) this pattern is less pronounced. This depth interval contributes the least to the total soil C mass, and as the model estimate is still not significantly different from the observations, we chose not to discuss this further in the paper.

**Referee comment:**

Line 288: How is 84% mean? Is it SOC/(SOC+Litter)? What did you mean by "the protected pools"?

**Author's response:**

*This means 84% of all carbon in the system, i.e. all nine pools. The "protected pools" refers to the pools SOMp and SOMc (figures and mass balances are found in Appendix A), which is protected from microbial decomposition, physically (SOMp) and (bio)chemically (SOMc) in the same manner as the original MIMICS version presented by Wieder et al. 2015. The sentence will be clarified in the revised manuscript.*

**Changes in manuscript:**

Parantheses referring to Fig. 1 (model illustration) was added to the sentence (line 334-335).

**Referee comment:**

Line: 298-299: There should be some pre-conditions for this implication. At global scale, productivities in warm regions are larger than those in cold regions, while SOC are mainly stored in northern high latitude.

**Author's response:**

*We will specify this by changing the sentence to "For the focus region (boreal sites in Norway) of this study, total C (TOTC) is strongly correlated with both MAT and C_input (+0.56 and +0.82, respectively) indicating that higher productivity at warmer sites is an important control on total soil C in the simulations (Fig. 5)."*

**Changes in manuscript:**

Since the correlations changed in the simulation reruns, the sentence was changed to "For the focus region (boreal sites in Norway) of this study, total C (TOTC) is strongly correlated with both MAT and C_input (+0.49 and +0.65, respectively) indicating that higher plant productivity at warmer sites is an important control on total soil C in the simulations (Fig. 5)." (line 345)

**Referee comment:**

Line 365-369: suggest to add few sentences to discuss the responses if increase in plant production (also mean increased N uptake by the plants).

**Author's response:**

*Good point! We will add some discussion about this in the discussion section.*

**Changes in manuscript:**

A sentence was added to line (416-417), and also discussed in lines 577-590 in Sect. 4.4

**Referee comment:**

Line 395-408: as I mentioned above, should the lower C:N is because there is some N loss process missing? As you model provided reasonable C stock, but too low C:N, it means you have too much N accumulated in the soil.

**Author's response:**

*As mentioned in the response to your previous comment, the discrepancy between observed and modeled C:N ratio can have several causes. For the revised manuscript we will rerun the simulations with corrected C allocation to mycorrhiza (see our response to reviewer 2 under "Results"). We expect this to bring modeled C:N values closer to observed due to increased N competition and loss through mycorrhiza. Generally, we think that parameter choices rather than missing processes in the model could explain these discrepancies. We will address this in connection to the parameter analysis in the revised manuscript.*

**Changes in manuscript:**

The new simulations gave better C:N agreement with the observations, however there is still an underestimation of soil C:N ratio. Possible causes are discussed in the two last paragraphs of Sect. 4.4 (lines 577-590)

**Referee comment:**

The discussion part should be better organized. For example, 4.1 and 4.2 both have C pools. Maybe you can have more clear subsection tiles?

**Author's response:**

*In this paper we partly perform model-observation comparisons (Sections 3.1 & 4.1), and partly analyze the model result itself (Sections 3.2 and 4.2). We discussed several approaches to the*

*organization of the sections, and found the current structure to be the most suitable for the story(?). To make the structure clearer to the reader, we will add a brief explanation to the beginning of the results section, and also revise the subsection titles.*

**Changes in manuscript:**

The following subsection titles have been changed:

3.1, 4.1: "Comparison of modelled and empirical C and N stocks"

3.2, 4.2: "Modelled C pools in MIMICS+"

2.3.1, 3.3 and 4.3 "Comparison of climate gradient profiles"

We also removed the subsubsections 3.2.1 and 4.2.1 (Modelled correlations) to follow journal guidelines (cannot have 3.2.1 without 3.2.2).

**Referee comment:**

Section 4.5: what is the C:N set for fungi and bacteria? How many microbial groups are in the model? Could the too low C:N is due to the prescribed C:N value? The C:N for microbes should be variable, you may add one sentence to discuss this.

**Author's response:**

*Details about model structure (four microbial pools) and parameters (C:N ratio etc.) are found in Methods and Appendix A. We will add some discussion about the consequences of the low, constant, microbial C:N ratios used in the model, along with references to the text/Appendix where it's needed. We will also include a parameter analysis for different values of microbial C:N which will inform the discussion.*

**Changes in manuscript:**

Added a paragraph to Sect. 4.4 (line 577-581) discussing this.

**Referee 2:**

*Introduction:*

**Referee comment:**

Please describe the potential mechanisms/processes involved with EcM and AM based on previous research;

Please describe the potential improvements over the models in Sulman et al. (2019) and Baskaran et al. (2017) and distinguish your study with other similar ones.

**Author's response:**

*We will describe mycorrhizal processes in more detail, including the Gadgil and priming effects that are of key importance when mycorrhizal processes are included in soil models that includes both carbon and nitrogen. Through this we will make a stronger link to work presented by Sulman et al. (2019) and Baskaran et al. (2017). In contrast to Baskaran et al., (2017) MIMICS+ models the N cycle, and is vertically resolved, allowing better representation of microbial competition for nitrogen. Compared to Sulman et al (2019), MIMICS+ presents an alternative structure of pools and fluxes with particular focus on mechanisms that are important in boreal/cold climates.*

**Changes in manuscript:**

Mechanisms related to mycorrhizal processes are introduced in the second paragraph of the introduction (line 35-45), while differences and potential improvements of MIMICS+ compared to the other models are introduced in the 5[th] paragraph (line 72-78).

***Methods:***

**Referee comment:**

Sensitivity analysis of parameters to modeled soil carbon is routine for model development and therefore is necessary here.

**Author's response:**

*We agree that sensitivity analysis of parameters (especially poorly constrained ones) should be included. The revised manuscript will include sensitivity analysis for key parameters, such as e.g. CUE, C:N ratios and direct litter-to-SOM fractions.*

**Changes in manuscript:**

Section 2.1.5 was added, describing the sensitivity analysis. A figure showing the total C sensitivity to the parameter perturbations was added to the appendix (Fig C5). The result is presented in line 329-332 and the figure is also referred to in Sect. 4.1 (line 453), Sect. 4.2 (line 492) and Sect. 4.4 (line 583).

**Referee comment:**

Line 103: describe in full C11, C12, N11, N12.

**Author's response:**

*These fluxes will be described in more detail.*

**Changes in manuscript:**

A more detailed description was added (line 116-120)

**Referee comment:**

Lines 162-164: add details about how EcM produces enzymes and the mining algorithm (since it is the import part of this study); is there carbon cost for modeled nitrogen mining?

**Author's response:**

*These details are described in the detailed figures and tables in the Appendix. The flux C27 is the carbon cost for ECM (from ECM to SOMa). We will add a sentence with references to the appendix in the main text.*

**Changes in manuscript:**

The fluxes are described in more detail (lines 180-185). The arrow representing flux C27 are corrected in Figure 1 and A1. In the previous figure it was incorrectly drawn from the EcM box, when it should be from the incoming flux.

**Referee comment:**

Lines 180-184: do they occur at the same time or sequentially? Please clarify. This is very important for nutrient competition.

**Author's response:**

*Numerically, they happen in the sequence presented in the manuscript, but since the time step is so short (1h) the order is of limited importance. Our understanding is that the processes in boreal*

*forests are relatively slow, so that the residence times of the reservoirs are much longer than the 1 hour time step. Therefore the inaccuracy implied by the sequential approach is small. We will add a short note about this in the text.*

**Changes in manuscript:**

Line 201-204 describes our approach.

**Referee comment:**

Lines 195-201: please describe here how the parameters change with depth, in particularly the ones associated with EcM and AM.

**Author's response:**

*The mycorrhizal parameters like turnover, mining, N uptake etc. are not explicitly dependent on environmental variables (temperature and moisture) and thereby not depth dependent. We will explain this in more detail in the text. During the revision we found that the weighting with respect to the thickness of the soil vertical layers were not entirely correct for the N uptake parameters, and has been changed for the rerun of the model (See our response under the "Results" header).*

**Changes in manuscript:**

Lines 225-227 describes how temperature and moisture are vertically resolved and thereby affect some rate equations. It is also described how mycorrhizal N uptake are dependent on biomass and SOM in the soil layer.

**Referee comment:**

Lines 214-216: cannot follow this sentence. Please elaborate.

*The sentence will be rewritten to say "Due to computational resource limitations, we chose a subset of 50 representative sites (out of the 578) for the site simulations with CLM and MIMICS+. The remaining 528 sites was used for further comparison with the modeled carbon stocks." We hope that it is clarifying.*

**Changes in manuscript:**

The sentence was rewritten as stated (line 244-246)

**Results:**

**Referee comment:**

The poor simulation results in terms of soil carbon (site level one to one), and CN ratio is concerning. Please justify.

**Author's response:**

*As the model is intended to work on larger spatial scales within an ESM model, a good one-to-one match with specific sites is of less importance than being able to capture larger patterns in temperature and moisture. Norwegian forest soils are also very heterogeneous and local features could have affected measurements in a way that not necessarily represent the general pattern we want to capture in a model. However, we agree that the poor one-to-one match is unsatisfactory. A recently discovered error in CLM in the amount of C allocated to mycorrhizal uptake, thus affecting the forcing of the MIMICS+ model (in terms of how much and in what form carbon was transferred from the vegetation to the soil) might explain at least part of the discrepancy between modeled and observed soil carbon and CN. For the revised manuscript we therefore want to rerun our*

*simulation with the correct (and higher) C allocation to mycorrhiza (with this error in CLM the allocation was too low). The extra carbon is a relatively small fraction of the overall carbon input to the system, but will alter the nutrient competition between microbes and could have an impact on model performance.*

**Changes in manuscript:**

The rerun simulations did not improve the one-to-one correlations (which are now shown in Fig. 3). Line 302-304 address the issue, and how we think the model can still be useful on larger scales, where local features are less important. It is also mentioned in line 425-426, 445-447.

**Referee comment:**

Please show seasonal variation of microbial respiration for model quality judgment. It can be shown in appendix though.

**Author's response:**

*We agree with the point about seasonal variation of microbial respiration, and will include a figure in the appendix showing this.*

**Changes in manuscript:**

Figure C6 was added to the Appendix, and it is referred to in line 334-335.

**Referee comment:**

*In the main text please show site-to-site correlation in addition to the boxplot. It is more informative.*

**Author's response:**

*The site-to-site correlation will be presented in the main text, in addition to the boxplot.*

**Changes in manuscript:**

Site-to-site correlations are now moved from Appendix to Fig. 2d-e.

**List of changes not related to referee comments or model reruns:**

43-47: Rewritten for better flow

50: Removed "can"

55-58: Rewritten to clarify.

66-67: Clarified sentence

82-83: Rewritten for flow

105-106: Corrected terminology

169: EcM and AM are now introduced in introduction

172: removed paranthesis

Table 1: Removed quotation marks

252-254: Added sentence referring to Table A6 with CLM variables

309-310: Moved paranthesis

421-422: Restructured sentences

428: Word choice

448: Removed paranteses

467: Added "Table"

483: Corrected word

521: Removed "we"

538 & 542: Moved figures to supplementary material

563: Added "dominates"

568-572: Removed, as this was mainly a consequence of the too low C allocated to mycorrhiza from CLM.

595: Removed word

---

## Author Response (AR2)

Thank you for your comments and corrections. Below follows a point-by-point reply (in red) to the comments.

Re: section 2.3.1 ("Comparison of climate gradient profiles"): Your reasoning is sound, but the text is still confusing.
- "local climate forcing" in the added text confuses more than it helps. Consider replacing that sentence with just: "We split the dataset this way—OBS by observed climate and MIMICS+ and CLM by model forcing climate—because we are investigating sensitivities to temperature and precipitation. (Fig. S1 shows results of this analysis with all points classified according to their OBS climate.)"
We agree that your sentence is clearer. The sentence have been changed to your suggestion.

- Mention Fig. S1 again in Sect. 3.3
The first sentence in Sect. 3.3 now reads "In Fig. 6 the 50 sites have been divided into two subsets of 25 sites based on climate categories described in Sect. 2.3.1. (Fig. S1 shows the result of the division of sites based only on the on the observed climate.)"

- L269 in revised manuscript (not tracked-changes): "sited" should be "sites"
The typo is corrected.

An additional typo in L267 was also corrected.

In tracked-changes version:
- L117: "moves" should be "move"
- L118: "represent" should be "represents"
- L230: "with" should be "by"
- L417: "was" should be "were"
The words listed above have been corrected.

- L425-6: This could also be due to inaccuracies in the model climate forcings.
Good point. To address this, the sentence is changed to "However, both models showed poor one-to-one agreement with the observations (Fig. 2d--f), possibly due to local heterogeneity that is not captured by the models and/or inaccuracies in the model climate forcings."

Figures:
- Fig. 1: Font size in labels at top of figure needs to be larger (comparable to figure caption size).
The figure is updated with larger font size for the text on top of the figure.

- Fig. 2: Including a 1:1 line for subplots d-f would be helpful.
1:1 lines are added to the subplots, with an explanation in the figure caption.

- Figs. 3, 6, 7, A1, B1, B4: Figure label font sizes need to be larger.
Font sizes are increased. For B1, the total figure size is also increased.